

# Impact of mineral dust on shortwave and longwave radiation: evaluation of different vertically-resolved parameterizations in 1-D radiative transfer computations

María José Granados-Muñoz[1], Michael Sicard[1,2], Roberto Román[3], Jose Antonio Benavent-Oltra[4,5],

Rubén Barragán[1,2], Gerard Brogniez[6], Cyrielle Denjean[7,8], Marc Mallet[7], Paola Formenti[8], Benjamín

Torres[6,9] and Lucas Alados-Arboledas[4,5]

[1] Remote Sensing Laboratory / CommSensLab, Universitat Politècnica de Catalunya, Barcelona, 08034, Spain
[2] Ciències i Tecnologies de l'Espai - Centre de Recerca de l'Aeronàutica i de l'Espai / Institut d'Estudis Espacials de Catalunya
(CTE-CRAE / IEEC), Universitat Politècnica de Catalunya, Barcelona, 08034, Spain
[3] Grupo de Óptica Atmosférica (GOA), Universidad de Valladolid, Valladolid, Spain.
[4] Department of Applied Physics, University of Granada, 18071 Granada, Spain
[5] Andalusian Institute for Earth System Research (IISTA-CEAMA), University of Granada, Autonomous Government of
Andalusia, 18006 Granada, Spain
[6] Laboratoire d'Optique Atmosphérique, University of Lille 1, Villeneuve d'Ascq, France
[7] CNRM, Centre National de la Recherche Météorologique (UMR3589, CNRS, Météo-France), Toulouse, France
[8] LISA, UMR CNRS 7583, Université Paris Est Créteil et Université Paris Diderot, Institut Pierre-Simon Laplace, Créteil,
France
[9] GRASP-SAS, Remote sensing developments, LOA/Université Lille-1, Villeneuve d'Ascq, France

*Correspondence to*: Maria Jose Granados (maria.jose.granados@tsc.upc.edu)

**Abstract.**

Aerosol radiative properties are investigated in southeastern Spain during a dust event on June 16-17,

2013 in the framework of the ChArMEx/ADRIMED (Chemistry-Aerosol Mediterranean Experiment

/Aerosol Direct Radiative Impact on the regional climate in the MEDiterranean region) campaign. Particle

optical and microphysical properties from ground-based sun/sky photometer and lidar measurements, as

well as in situ measurements onboard the SAFIRE ATR 42 French research aircraft are used to create a

set of different levels of input parameterizations, which feed the 1-D radiative transfer model (RTM)

GAME (Global Atmospheric ModEl). We consider three datasets: 1) a first parameterization based on

the retrievals by an advanced aerosol inversion code (GRASP; Generalized Retrieval of Aerosol and

Surface Properties) applied to combined photometer and lidar data; 2) a parameterization based on the

photometer columnar optical properties and vertically-resolved lidar retrievals with the two-component



Klett-Fernald algorithm; and 3) a parameterization based on vertically-resolved optical and microphysical aerosol properties measured in situ by the aircraft instrumentation. Once retrieved, the outputs of the RTM in terms of both shortwave and longwave radiative fluxes are contrasted against ground-, satellite- and in situ airborne measurements. In addition, the outputs of the model in terms of the aerosol direct radiative effect are discussed with respect to the different input parameterizations. Results show that calculated atmospheric radiative fluxes differ no more than 7 % to the measured ones. The three parameterization datasets produce aerosol radiative effects with differences up to 10 $W \cdot m^{-2}$ in the shortwave spectral range (mostly due to differences in the aerosol optical depth), and 2 $W \cdot m^{-2}$ for the longwave (mainly due to differences in the aerosol optical depth but also to the coarse mode radius used to calculate the radiative properties). The study reveals the complexity of parametrizing 1-D RTMs as sizing and characterising the optical properties of mineral dust is challenging. The use of advanced remote sensing data and processing, in combination with closure studies on the optical/microphysical properties from in situ aircraft measurements when available, is recommended.

## 1 Introduction

The radiative effect by atmospheric aerosol is estimated to produce a net cooling effect of the Earth's climate. However, an accurate quantification of this cooling is extremely difficult. In fact, the aerosol radiative effect (ARE) is affected by large uncertainties. Due to the direct aerosol-radiation interaction, the ARE is estimated to be $-0.27$ $W \cdot m^{-2}$ on average at the global scale, with an uncertainty range of $-0.77$ to $-0.23$ $W \cdot m^{-2}$, whereas the radiative effect related to cloud adjustments due to aerosols is $-0.55$ $W \cdot m^{-2}$ ($-1.33$ to $-0.06$ $W \cdot m^{-2}$) (Boucher *et al.*, 2013), being the largest unknown in the radiative forcing of the atmosphere. The extent to which the ARE uncertainty range reported is physical or due to measurement artefacts is still hard to quantify.

In previous studies, the aerosol radiative effects in the longwave (LW) were commonly neglected due to the complexity of an accurate quantification of the optical properties in this spectral range (Roger et al., 2006; Mallet et al., 2008; Sicard et al., 2012). However, the contribution of the LW component to the ARE is non-negligible for large aerosol particles, i.e., marine aerosol or mineral dust (e.g. Markowicz



et al., 2003; Vogelmann et al., 2003; Otto et al., 2007; Perrone and Bergamo, 2011; Sicard et al., 2014a,b; Meloni et al., 2018).

The contribution of mineral dust to the ARE in the infrared spectral range is especially relevant because of its large size and abundance (Meloni et al., 2018). Mineral dust is estimated to be the most abundant aerosol type in the atmosphere by mass (e.g., Ginoux et al., 2012; Choobari et al., 2014), with global emission between 1000 and 3000 Mt·yr$^{-1}$ (Zender et al., 2003; 2004; Shao et al., 2011). The high temporal and spatial variability of dust concentrations and the variability in their microphysical and optical properties present a significant challenge to our understanding of how these particles impact the environment (Dubovik et al., 2002). Many measurements worldwide have been made using different approaches, including satellites which can provide global coverage of mineral dust properties. However, the retrievals of particle properties are still affected by large uncertainties (Levy et al., 2013) and the information on mineral dust properties is quite scarce (Formenti et al., 2011).

One of the areas frequently influenced by mineral dust is the Mediterranean Sea region, affected by dust intrusions from the close Sahara Desert or the Middle-East region (Moulin et al., 1998; Israelevich et al., 2012; Gkikas et al., 2013) producing significant perturbations to the shortwave (SW) and the LW radiation balance (di Sarra et al. 2011; Meloni et al. 2015; Perrone et al., 2012) as well as the regional climate (Nabat et al., 2015).

To address these issues, the Aerosol Direct Radiative Impact on the regional climate in the MEDiterranean region (ADRIMED) field campaign within the Chemistry-Aerosol Mediterranean Experiment (ChArMEx, http://charmex.lsce.ipsl.fr) took place in the Mediterranean region from 11 June to 5 July 2013 (Mallet et al., 2016). It aimed at characterizing the different aerosol particles and their radiative effects using airborne and ground-based measurements collected in the Mediterranean Basin, with special focus on the western region. In particular, two ChArMEx/ADRIMED flights, F30 and F31, from the French ATR 42 environmental research aircraft of SAFIRE (http://www.SAFIRE.fr), took place above southeastern Spain during a Saharan dust episode on 16 and 17 June 2013.

In this paper, we present an analysis of the mineral dust radiative properties during this particular episode taking advantage of the thorough database available. Multiple datasets are used as input in a radiative transfer model (RTM) to evaluate the influence of the different measurements and data



processing in the retrieved direct ARE. The model used here is the Global Atmospheric ModEl (GAME; Dubuisson et al., 1996; 2005), which allows calculating both the solar and thermal infrared fluxes. An evaluation against aircraft in situ measurements of radiative fluxes is also presented.

Two main goals are pursued: i) the quantification of the direct ARE for two case studies within a dust transport episode and ii) the evaluation of the model estimates sensitivity to the aerosol input used.

The paper is structured as follows: Section 2 includes a description of both the ground-based and in situ aircraft instrumentation and a short description of the retrieval algorithms used for the present study; Section 3 is devoted to the description of GAME and the input datasets used here and results are presented in Section 4; finally, a short summary and concluding remarks are included in Section 5.

## 2 Instruments and data

### 2.1 Ground-based measurements

Ground-based measurements used in this work were carried out at the Andalusian Institute for Earth System Research (IISTA-CEAMA) of the University of Granada, Spain (37.16º N, 3.61º W, 680 m a.s.l.) by the Atmospheric Physics Group of the University of Granada (GFAT-UGR). This experimental site is located in the western Mediterranean basin, near the African continent (~200 km). Therefore, long-range transport of mineral dust particles from North Africa is a main source of natural atmospheric aerosol in the region (e.g. Lyamani et al., 2005; Valenzuela et al., 2012). The station is also affected by long-range transported smoke (Ortiz-Amezcua et al., 2017) and fresh smoke from nearby biomass burning (Alados-Arboledas et al., 2011). Anthropogenic sources such as pollution from Europe, the Iberian Peninsula and the Mediterranean Sea (Pérez-Ramírez et al., 2016) also affect the station. Local sources are mainly road traffic and central heating systems (Titos et al., 2017).

IISTA-CEAMA station is equipped with a CE-318-4 (*Cimel Electronique*) sun/sky photometer which belongs to the AERONET network (Holben et al., 1998). This instrument makes direct solar irradiance measurements, used to derive aerosol optical depth (AOD), and sky radiance measurements both at least at the following nominal wavelengths ($\lambda$): 440, 670, 870 and 1020 nm. The AOD product provided by AERONET have an uncertainty of ±0.01 for $\lambda > 440$ nm and of ±0.02 for $\lambda < 440$ nm (Holben



et al., 1998; Eck et al., 1999). AERONET also provides aerosol optical and microphysical properties such as columnar particle size distribution (PSD), real and imaginary refractive indices (RRI and IRI, respectively), asymmetry factor ($g$) and single scattering albedo (SSA), using the AOD and sky radiance values in an inversion algorithm (Dubovik and King, 2000; Dubovik et al., 2006). For the present study,

AERONET Version 2 Level 1.5 (Level 2.0 when available) data are used. The uncertainty in the retrieval of SSA is ±0.03 for high aerosol load ($AOD_{440} > 0.4$) and solar zenith angle $> 50º$; while for measurements with low aerosol load ($AOD_{440} < 0.2$), the retrieval accuracy of SSA drops down to 0.02–0.07 (Dubovik and King, 2000). For high aerosol load and solar zenith angle $> 50°$, errors are about $30\%-50\%$ for the IRI. For particles in the size range $0.1 < r < 7$ μm (being r the aerosol radius), errors in PSD retrievals are

around 10–35%, while for sizes lower than 1 μm and higher than 7 μm retrieval errors rise up to 80–100%. The inversion code provides additional variables such as the volume concentration, effective radius, $r_{eff}$, and geometric standard deviation of the equivalent lognormal distribution, σ, for fine and coarse modes of the retrieved PSD which will be used in the current study.

The multi-wavelength aerosol Raman lidar MULHACEN, based on a customized version of

LR331D400 (Raymetrics S.A.), is operated at Granada station as part of EARLINET/ACTRIS (European Aerosol Research Lidar Network / Aerosols, Clouds, and Trace Gases Research Infrastructure Network; https://www.actris.eu/default.aspx; Pappalardo et al., 2014) since April 2005. The system has a monostatic biaxial configuration, which usually requires an overlap correction to minimize the incomplete overlap effect (Navas-Guzmán et al., 2011). The system emits vertically to the zenith by means of a pulsed

Nd:YAG laser with 2nd- and 3rd-harmonic generators, that emits simultaneously at 1064, 532 and 355 nm. The receiving system consists of several detectors, which can split the radiation according to the three elastic channels at 355, 532 (parallel- and perpendicular-polarized; Bravo-Aranda et al., 2013), and at 1064 nm; two nitrogen Raman channels at 387 and 607 nm; and a water vapor Raman channel at 408 nm (Navas-Guzmán et al., 2014). The aerosol particle backscatter coefficient profiles ($β_{aer}(z,λ)$, being z the

vertical height) obtained from the multi-wavelength lidar were calculated with the Klett-Fernald method (Fernald et al., 1972; Fernald, 1984; Klett, 1981, 1985). For the retrieval of the aerosol extinction coefficient profiles ($α_{aer}(z,λ)$), a height-independent lidar ratio (LR) obtained by forcing the spatial integral of $α_{aer}(z,λ)$ to the AOD from AERONET photometer (Landulfo et al., 2003) was assumed. The



assumption of a constant LR introduces uncertainty in $\alpha_{aer}(z,\lambda)$ retrievals, especially when different types of aerosol appear at different layers. In our case, the LR used for the Klett-Fernald retrieval are very similar to those provided by GRASP (see Benavent-Oltra et al., 2017). Considering the different uncertainty sources, total uncertainty in the profiles obtained with Klett-Fernald method is usually 20%

for $\beta_{aer}(z,\lambda)$ and 25-30% for $\alpha_{aer}(z,\lambda)$ profiles (Franke et al., 2001).

Additionally, surface temperature and pressure are continuously monitored at IISTA-CEAMA in a meteorological station located 2 m above the ground. At the same location, the global and diffuse downward radiative fluxes for the SW are continuously measured with a CM11 pyranometer (Kipp & Zonen) and diffuse downward radiative fluxes for the LW are measured with a PIR pyrgeometer (Eppley),

being both instruments regularly calibrated at the site (Antón et al., 2012, 2014).

## 2.2 Airborne measurements

The Safire ATR 42 aircraft performed two overpasses above Granada on June 16 (flight F30) and 17 (flight F31) in 2013 during the ChArMEx/ADRIMED campaign. During F30, the SAFIRE ATR 42 descended performing a spiral trajectory from 14:15 to 14:45 UTC, whereas during flight F31, the aircraft

ascended in the early morning (from 07:15 to 07:45 UTC) at around 20 km from Granada station (see Fig. 1 from Benavent-Oltra et al., 2017). Additional flight details can be found in previous studies (Denjean et al., 2016; Mallet et al., 2016; Benavent-oltra et al., 2017; Román et al., 2018).

The airborne instrumentation includes a Scanning Mobility Particle Sizer (SMPS) and an Ultra-High Sensitivity Aerosol Spectrometer (UHSAS), for measuring aerosol number size distribution in the

submicron range. The Forward Scattering Spectrometer Probe model 300 (FSSP-300) and the GRIMM OPC (sky-OPC 1.129) were used to measure the optical size distributions in the diameter nominal size range between 0.28 and 20 µm and between 0.3 and 32 µm, respectively. A nephelometer (TSI Inc, model 3563) was used to measure the particle scattering coefficient at 450, 550 and 700 nm, and a Cavity Attenuated Phase Shift (CAPS-PMex, Aerodyne Inc.), was employed to obtain the aerosol extinction

coefficient ($\alpha_{aer}$) at 530 nm. For more details on the aircraft instrumentation see Denjean et al. (2016) and references therein. The PLASMA (Photomètre Léger Aéroporté pour la Surveillance des Masses d'Air) system, which is an airborne sun-tracking photometer, was additionally used to obtain AOD with wide



spectral coverage (15 channels between 0.34 – 2.25 µm) with an accuracy of approximately 0.01, as well as the vertical profiles of the aerosol extinction coefficient (Karol et al., 2013; Torres et al., 2017).

Airborne radiative fluxes (F) were measured with Kipp & Zonen CMP22 pyranometers and CGR4 pyrgeometers. Upward and downward SW fluxes ($^{\uparrow}F_{SW}$ and $^{\downarrow}F_{SW}$) were measured in the spectral range

297-3100 nm by two instruments located above and below the aircraft fuselage. The same setup was used for the pyrgeometers, which provided the LW upward and downward radiative fluxes ($^{\uparrow}F_{LW}$ and $^{\downarrow}F_{LW}$) for wavelengths larger than 4 µm. Both pyranometers and pyrgeometers were calibrated in January 2013 and data were corrected for the temperature dependence of the radiometer's sensitivity following Saunders et al. (1992).

Downward pyrgeometer measurements were filtered out for large pitch and roll angles and corrected from the rapid variations of the solar incidence angle around the solar zenith angle due to the aircraft attitude (pitch and roll). This correction also depends on aircraft heading angle and solar position. It should be noted that, beforehand, roll and pitch offsets must be determined (the axis sensor is not necessarily vertical on average during a horizontal leg). Cosine errors were taken into account. Finally,

data were corrected from variations of the solar zenith angle (SZA) during the flight to ease the comparison with GAME retrievals. After these various corrections, an estimated uncertainty of $\pm 6$ W·m$^{-2}$ is considered to affect the data, taking into account the accuracy of the calibration and of the acquisition system together with the consistency of airborne measurements (Meloni et al., 2018).

## 2.3. The GRASP code

The GRASP (Generalized Retrieval of Aerosol and Surface Properties) code (Dubovik et al., 2011, 2014), provides aerosol optical and microphysical properties in the atmosphere by combining the information from a variety of remote sensors (e.g. Kokhanovsky et al., 2015; Espinosa et al., 2017; Torres et al., 2017; Román et al., 2017, 2018; Chen et al., in review). In our case, GRASP was used to invert simultaneously coincident lidar data (range corrected signal, RCS, at 355, 532 and 1064 nm) and sun/sky photometer

measurements (AOD and sky radiances both at 440, 675, 870 and 1020 nm) providing a detailed characterization of the aerosol properties, both column-integrated and vertically-resolved. It is worthy to note that this GRASP scheme, based on Lopatin et al. (2013), presents the main advantage that it allows



retrieving aerosol optical and microphysical properties for two distinct aerosol modes, namely fine and coarse. The $\alpha_{aer}$, $\beta_{aer}$, SSA (all at 355, 440, 532, 675, 870, 1020 and 1064 nm), and aerosol volume concentration (VC) profiles obtained as output from GRASP will be used as input to GAME in the present study, together with the column-integrated PSD properties (namely $r_{eff}$ and σ for fine and coarse modes). A more in-depth analysis of GRASP output data retrieved using the lidar and sun/sky photometer data at Granada station for the two inversions simultaneous to the aircraft overpasses during flights F30 and F31 during ChArMEx/ADRIMED campaign can be found in Benavent-Oltra et al. (2017).

## 3. GAME radiative transfer model

### 3.1. GAME description

The GAME code is widely described by Dubuisson et al. (2004; 2005) and Sicard et al. (2014a). It is a modular RTM that allows calculating upward and downward radiative fluxes at different vertical levels from the ground up to 20 km (100 km) in the SW (LW) spectral range. The solar and thermal infrared fluxes are calculated in two adjustable spectral ranges, which in this study were fixed to match those of the aircraft radiation measurements, namely 297 - 3100 nm for the SW and 4.5 – 40 μm for the LW, by using the discrete ordinates method (Stamnes et al., 1988). Note that the GAME code has a variable spectral sampling in the SW (depending on the spectral range considered) and a fixed spectral sampling (115 values) in the LW spectral range (Table 1).

[Table 1]

### 3.2. GAME input data parameterization

The two considered SAFIRE ATR 42 flights, F30 and F31, took place on 16 and 17 June 2013, respectively, simultaneously to ground-based lidar and sun/sky photometer measurements performed at the station. On these days, mineral dust with origin in the Sahara region (southern Morocco near the border with Algeria) reached Granada after ~4 days of travelling, according to back-trajectories analysis (not shown) and the results presented in Denjean et al. (2016). A homogenous dust layer reaching up to 5 km agl was observed on June 16, whereas on June 17 the dust layer was decoupled from the boundary layer and located between 2 and 4.5 km agl (Benavent-Oltra et al., 2017). On June 16, the F30 profile



took place between 14:15 and 14:45 UTC (averaged SZA=31.49º) in coincidence with the lidar measurements. The sun/sky photometer microphysics data were not available till 16:22 UTC, even though the retrieved AOD and its spectral dependence (represented by the Angström exponent) were very stable between the time of the lidar measurements and the time of the sun/sky photometer inversion. On June

17, the F31 profile occurred in the early morning (07:15 to 07:45 UTC, averaged SZA=61.93º), and simultaneous lidar and sun/sky photometer were available. Unfortunately, the airborne vertical profile of extinction by the CAPS measurements was not available during this second flight. Clouds were detected by the lidar on June 17 after 15:00 UTC. Furthermore, the ground-based pyranometer and pyrgeometer data indicate cloud contamination in the radiation data much earlier (around 09:00 UTC), preventing also

satellite retrievals in the region.

A summary of the experimental data used as input for GAME calculations during these two case studies is presented in Table 2. This input includes namely surface parameters and atmospheric profiles of meteorological variables, main gases concentrations and aerosol properties. The aerosol properties used in the present study are parameterized using three different datasets, based on the different instrumentation

and retrievals available, i.e. Dataset 1 (DS1), Dataset 2 (DS2) and Dataset 3 (DS3). A more detailed description of the different parameters is provided next.

[Table 2]

### 3.2.1. Surface parameters and profiles of meteorological variables

The surface parameters required for GAME are the surface albedo ($alb(\lambda)$) and land-surface temperature (LST). The $alb(\lambda)$ for the SW range is obtained from the sun/sky photometer data using the AERONET retrieval at 440, 675, 880 and 1020 nm, and for the LW from the emissivity provided by the Single Scanner Footprint (SSF) Level2 products of the CERES (Clouds and the Earth's Radiant Energy System; (http://ceres.larc.nasa.gov/) instrument (Table 3). LST values are obtained from MODIS (Moderate

Resolution Imaging Spectroradiometer) 1-km daily level-3 data (Wan et al., 2014) on June 16. Unfortunately, on June 17 MODIS data were not available due to the presence of clouds and the local surface temperature was estimated from temperature measurements at Granada site, where the



meteorological station is located at 2 m above the ground. LST and $alb(\lambda)$ values used for the two analyzed cases are included in Table 3.

[Table 3]

Figure 1 shows the pressure (P), temperature (T), and relative humidity (RH) profiles obtained from the SAFIRE ATR 42 measurements. Data from the meteorological station located at IISTA-CEAMA are used to complete these profiles at the surface level, whereas at altitudes above the aircraft flight, a scaled US standard atmosphere is used for completion. The concentration profiles of the main absorbing gases ($O_3$, $CH_4$, $N_2O$, CO and $CO_2$) are also taken from the US standard atmosphere, while for the gaseous absorption coefficients the HITRAN database is used.

[Figure 1]

### 3.2.2. Aerosol parameterization

As for the aerosol parameterization, $\alpha_{aer}(\lambda,z)$, $SSA(\lambda,z)$ and $g(\lambda,z)$ are required as GAME input data (Table 2). For the SW wavelengths, these properties can be obtained from the measurements performed with the instrumentation available during the campaign; namely the lidar, the sun/sky photometer and the in situ instrumentation onboard the aircraft. On the other hand, direct measurements of the aerosol properties in the LW are not so straightforward and thus scarce. Hence, the aerosol LW radiative properties are calculated by a Mie code included as a module in GAME. According to Yang et al. (2007), the dust particles non-sphericity effect at the thermal infrared wavelengths is not significant on the LW direct ARE, thus the shape of the mineral dust can be assumed as spherical for the Mie code retrievals introducing negligible uncertainties.

For the SW simulations, we run GAME using three different aerosol input datasets, i.e. DS1, DS2 and DS3 (Table 2), in order to evaluate their influence on the ARE calculations. DS1 relies on a parameterization based on the advanced post-processing GRASP code, which combines lidar and sun/sky photometer data to retrieve aerosol optical and microphysical properties profiles; DS2 relies on Klett-Fernald lidar inversions and AERONET products and corresponds to a reference parameterization (easily reproducible at any station equipped with a single- or multi-wavelength lidar and an AERONET sun/sky photometer and without the need of an advanced post-processing algorithm); and DS3 relies on in situ airborne measurements and corresponds to an alternative parameterization to DS1 and DS2.



Figure 2 shows $\alpha_{aer}$ profiles on June 16 (top) and 17 (bottom) obtained using the three different approaches. For DS1 (Figure 2a and d), $\alpha_{aer}$ profiles at seven different wavelengths obtained with GRASP are used as input data in GAME. In DS2 (Figure 2b and 3e), the $\alpha_{aer}$ profiles are obtained from the lidar data using Klett-Fernald retrievals and adjusting the lidar ratio to the AERONET retrieved AODs, as

mentioned in Section 2.1. Finally, for DS3 the $\alpha_{aer}$ values are obtained from the aircraft in situ measurements (CAPS and PLASMA data on June 16 and PLASMA on June 17). A detailed analysis and discussion on the comparison between $\alpha_{aer}$ profiles provided by the aircraft measurements, GRASP and the lidar system at Granada is already included in Benavent-Oltra et al. (2017). In general, the lidar, GRASP and the CAPS data are in accordance, observing the same aerosol layers and similar values, with

discrepancies within 20%. Considering that the uncertainty in $\alpha_{aer}$ is around 30% for both GRASP and the Klett-Fernald retrieval and 3% for the CAPS data, this discrepancy is well below the combined uncertainty of the different datasets. Discrepancies in the $\alpha_{aer}$ profiles translate into differences in the integrated extinction and, hence, in differences in the AOD values used as input in the radiative fluxes retrievals. The AOD values presented here (included in Table 4) are obtained by integrating the $\alpha_{aer}$

profiles, interpolated at 550 nm, from the surface up to the top of the aerosol layer (4.3 km on June 16 and 4.7 km on June 17). The AOD values at 550 nm reveal that GRASP input data (DS1) and in a lesser extent the aircraft in situ data (DS3) underestimate the aerosol load in the analyzed dust layer compared to AERONET (DS2) due to the differences in the retrieval techniques, e.g. whereas AERONET provides integrated AOD for the whole column, low $\alpha_{aer}$ values above the aerosol layer are neglected for the AOD

calculations in DS1 and DS3.

[Figure 2]

[Table 4]



Figure 3 presents the SSA values retrieved by GRASP algorithm, used as input for GAME in DS1, on June 16 (F30, Figure 3a) and 17 (F31, Figure 3b). On June 17 the SSA profiles present lower values and more variation with height than on June 16; the lower SSA values indicate the presence of more absorbing particles on June 17. For DS2, the SSA are taken from AERONET columnar values and assumed to be

constant with height (Figure 4a). As in Figure 3, SSA values are lower on June 17 due to the intrusion of more absorbing particles. For DS3, SSA values at 530 nm are obtained from the nephelometer and the CAPS or PLASMA onboard the ATR. In order to reduce the uncertainty of the measured data, only averaged values for the column will be considered, being 0.88 and 0.83 on June 16 and June 17 (Figure 4). Despite the difference between the aircraft and AERONET SSA values, the retrieved SSA values

obtained here are within the range of typical values for dust aerosols (Dubovik et al., 2002; Lopatin et al., 2013) and discrepancies are still within the uncertainty limits, which range between 0.02 and 0.07 depending on the aerosol load for AERONET data (Dubovik et al., 2000) and is 0.04 for the aircraft values. In the case of $g$ values, the same data are used for the three aerosol input datasets. Multispectral values of $g$ are taken from AERONET columnar values and assumed to be constant with height (Figure

4b).

[Figure 3]

[Figure 4]



For the LW calculations, the Mie code is used to obtain $\alpha_{aer}(\lambda,z)$, $SSA(\lambda,z)$ and $g(\lambda,z)$ from the information on the aerosol PSD, complex refractive index (RI) and density. A summary of the aerosol parameters used in the Mie calculations is included in Table 5. Three different datasets are also used for

the aerosol parameterization in the LW calculations. In this case, the sensitivity of the model to the PSD used is tested. A similar scheme to that presented for the SW is used, where DS1 relies on GRASP retrievals, DS2 on AERONET products and DS3 relies on in situ airborne measurements.

[Table 5]

The spectral real and imaginary parts of the RI of mineral dust in the LW are obtained from Di Biagio et

al. (2017), using the Morocco source, and assumed constant with height. The analysis by Di Biagio et al. (2017) only covers the spectral range 3-16 μm, so an interpolation assuming the spectral dependence presented in Krekov (1993) for shorter wavelengths is performed. The mineral dust particle density is assumed to be 2.6 g·cm$^{-3}$ (Hess et al., 1998). Regarding the PSD, three parameters (namely the effective radii, $r_{eff}$, standard deviation, σ, and the numeric concentrations, $N$) for fine and coarse modes are used.

The fine mode comprises particles within the diameter range 0.1−1 μm, whereas for the coarse mode the range 1-30 μm is considered. In the case of DS1, $N$ values are obtained from the volume concentration profiles provided by GRASP assuming spherical particles in the range between 0.05 and 15 μm radii (Figure 5). Values of $r_{eff}$ and σ provided by GRASP (Table 4) are column-integrated and thus assumed to be constant with height. This is the case also for DS2, in which the PSD parameters are column-integrated

values provided by the AERONET retrieval in Granada (see Table 4).

[Figure 5]

For DS3, the volume concentration (or the equivalent $N$), $r$ and σ profiles for the fine and coarse modes (Figure 5) are calculated from the data provided by the aircraft in situ measurements in the range between 0.02 and 40 μm diameter. Benavent-Oltra et al. (2017) found a general good accordance between the

volume concentration profiles measured by the instrumentation onboard the SAFIRE ATR 42 and retrieved with GRASP, with discrepancies within the combined uncertainty. Nonetheless differences are still noticeable, especially in the fine mode. On June 17 GRASP overestimates the aircraft measurements for the fine mode and underestimates them for the coarse mode, which in turns results in a quite different





fine to coarse concentration ratio for DS1 and DS3. Differences are mostly technical, i.e., GRASP retrieval is based on 30-min averaged lidar profiles while the aircraft provide instantaneous measurements, but they can be also partially caused by the discrepancies between the vertical aerosol distribution above Granada (sampled by the lidar) and the concentration measured during the aircraft

trajectory as they are not exactly coincident. In addition, for June 16, there is a 2 hours' time difference between the sun/sky photometer retrieval used in GRASP calculations and the airborne measurements which can lead to slight differences in the aerosol properties despite the homogeneity of the dust event during this period. In the following, we quantify the impact these differences may introduce in the calculations of F.

### 3.2.3. GAME output data

As a result of the simulation, GAME provides vertical profiles of radiative fluxes in the visible ($F_{SW}$) and thermal ($F_{LW}$) spectral ranges. The net flux can be calculated from the obtained profiles for both spectral ranges as:

$$\text{Net } F = {^\downarrow}F - {^\uparrow}F \hspace{4cm} \textbf{Equation 1}$$

where the upward and downward arrows are for upward and downward fluxes respectively. From the obtained radiative fluxes profiles, the direct ARE profiles are calculated according to the following equation:

$$ARE = ({^\downarrow}F^w - {^\uparrow}F^w) - ({^\downarrow}F^o - {^\uparrow}F^o) \hspace{3cm} \textbf{Equation 2}$$

where $F^w$ and $F^o$ are the radiative fluxes with and without aerosols, respectively. The direct ARE can be

obtained for the SW ($ARE_{SW}$) and the LW ($ARE_{LW}$) spectral ranges.

## 4 Mineral dust effect on shortwave and longwave radiation

### 4.1. SW radiative fluxes

Figure 6 shows the radiative fluxes profiles for the SW spectral range obtained with GAME using the

three different input datasets described in Section 3, as well as the Net $F_{SW}$. The radiative fluxes measured by the pyranometer onboard the SAFIRE ATR 42 are also included in the figure. The three GAME simulations show similar values with discrepancies below 8 W·m$^{-2}$ on average, which represents less than



1% variation. The differences in the obtained fluxes are mostly due to the differences in the aerosol load considered depending on the inputs. Even though the discrepancies in the AOD are within the uncertainty considered for the different datasets, they can lead to differences in $F_{SW}$ and ultimately in the $ARE_{SW}$. The larger AOD assumed for DS2 on both days (see Table 4 and Figure 2), causes the $^{\downarrow}F_{SW}$ to be slightly

lower compared to DS1. For DS3 we can also observe the effect of the SSA values used, which are relatively smaller compared to those measured by AERONET (see Figure 4), and lead to lower values of the radiative fluxes, despite the AOD being larger than for DS1.

  The evaluation against the aircraft measurements is limited to the altitude range below 5 km but this does not present a limitation since most of aerosols are below that height (see Figure 2 and Figure 5).

Larger discrepancies are observed for altitudes below 2.5 km (~860 mbar) on June 16, whereas a better agreement is found above. This is somehow expected due to the distance between the flight trajectory and the ground-based station (~20 km) and the influence of the boundary layer in the lower altitude range. On June 17, no $^{\uparrow}F_{SW}$ aircraft data are available above 2 km. Differences between the model and the aircraft measured data are well below 7%, being the largest discrepancies observed for the $^{\downarrow}F_{SW}$. Discrepancies

between the three GAME outputs and the aircraft pyranometer are lower than 5% for the Net $F_{SW}$ on both days. Considering the very different approaches followed by the model and the direct measurements by the airborne pyranometer, the differences are quite insignificant and a conclusive result on which input dataset provides a better performance is unlikely.

[Figure 6]

The values at the surface (or bottom of the atmosphere, BOA) and at the top of the atmosphere (TOA) for the different radiative fluxes can be also evaluated against different instruments: measurements for the $^{\downarrow}F_{SW}$ at the surface are available from the CM11 pyranometer located at the ground-station in Granada; CERES provides data of the $^{\uparrow}F_{SW}$ at the TOA; and AERONET provides values of the $^{\downarrow}F_{SW}$ and $^{\uparrow}F_{SW}$ at both the BOA and the TOA. The time series for these measurements corresponding to 16-17 June

and the results obtained with GAME for the different datasets are shown in Figure 7. Agreement is found between AERONET, the CM11 and the three GAME simulations on June 17 at the BOA. On June 16, when the radiation presents larger values, a 6% overestimation is observed in GAME (around 60 W·m$^{-2}$) compared to the pyranometer. A direct comparison with AERONET values is not possible on June 16



since GAME retrievals and the sun/sky photometer measurements are ~2 hours away. At the TOA, the $^{\uparrow}F_{SW}$ between GAME and AERONET are in agreement on June 17, when measurements are coincident. In the case of CERES, only a qualitative comparison is possible since no simultaneous and co-located data are available for the analyzed dates. CERES overpasses Southern Spain during daytime at $10 - 11$ UTC in summer while the flights were at 14:30UTC and 07:30UTC on June 16 and 17, respectively, so that both datasets are not simultaneous. As an example, Sicard et al. (2016) compared AERONET to CERES $F_{SW}$ at the TOA, and allowed a time difference between AERONET and CERES within ±15 min for validation purposes. Since no closest data were available on 16 and 17 June, we selected satellite overpasses within 600 km from Granada, thus differences are expected. It is also worth noticing that, as reported by Sicard et al. (2016), CERES fluxes present two major sources of artificially added flux: 1) the presence of clouds in the pixel, which is very likely here on 17 June, and 2) possible sun-glints. In both cases the result is an increase of CERES upward fluxes at the TOA. In addition, while the spectral range of GAME was set to the aircraft SW band ($0.297 - 3.1$ μm), the shortwave window of CERES is $0 - 5$ μm. Thus, even though data are consistent, the comparison of CERES with GAME simulations cannot be considered as conclusive.

[Figure 7]

The $ARE_{SW}$ profiles, calculated by using Eq. 2 and GAME simulations for the three input datasets are shown in Figure 8, together with the simultaneous values provided by AERONET on 17 June at the BOA and TOA. Comparing the three GAME simulations, we can see that the low discrepancies in the F profiles from Figure 6 lead to variations in the $ARE_{SW}$ of 10-27% over the averaged profile depending on the input dataset used. The variations in the $ARE_{SW}$ are tightly connected to discrepancies in the AOD considered as input in the model, as already observed in previous studies (Sicard et al., 2014a; Lolli et al., 2018; Meloni et al., 2018). The SSA and the vertical distribution of the aerosol also plays an important role, as observed for DS3, which shows a profile quite different from DS2 despite the AOD being quite close for both datasets.

Differences are also observed when comparing $ARE_{SW}$ values obtained from GAME to those retrieved by AERONET. Contrary to GAME simulations, AERONET does not consider the vertical distribution of the aerosols when calculating the $ARE_{SW}$, and the definition of the $ARE_{SW}$ at the BOA



($^{BOA}ARE_{SW}$) is slightly different. Indeed, AERONET $^{BOA}ARE_{SW}$ is calculated as the difference between the downward fluxes with and without aerosols, the difference between the upward fluxes (reflected by the Earth) being neglected. Considering this, we can correct the $^{BOA}ARE_{SW}$ provided by AERONET multiplying by a factor (1- $alb(\lambda)$). The corrected $^{BOA}ARE_{SW}$ value on 17 June is thus -31.9 W·m$^{-2}$, which is within the range of values provided by GAME at the surface. All discrepancies observed here are mostly intrinsic to the different techniques used for the acquisition of the data and the retrieval algorithms. The effect of the data processing has also been observed in previous studies (Lolli et al., 2018).

The $ARE_{SW}$ values obtained at the BOA and TOA for the three datasets and the averaged value are included in Table 6. Both at the BOA and TOA, the $ARE_{SW}$ has a cooling effect, as expected for mineral dust in this region according to values obtained in the literature (e.g. Sicard et al. 2014a, Mallet et al., 2016). The values obtained for this case are very similar to the regional summer mean value obtained by Papadimas et al., (2012) at the surface (-26.5 W·m$^{-2}$) and the TOA (-6.3 W·m$^{-2}$) and within the range of previous values observed in the western Mediterranean region, which varied from −93.1 to −0.5 W·m$^{-2}$ at the BOA and -34.5 to +8.5 W·m$^{-2}$ at the TOA (Sicard et al., 2014a,b; Barragan et al., 2017).

[Table 6]

[Figure 8]

## 4.2. LW radiative fluxes

Figure 9 shows $F_{LW}$ calculated with GAME after obtaining the aerosol properties in the LW spectral range from Mie calculations from the three mentioned datasets (see Section 3.2.2). $F_{LW}$ measured by pyrgeometers located onboard the ATR is also shown.

In general, differences in the $F_{LW}$ are always lower than 6%, with the airborne values being overestimated by the model on 16 June and underestimated on 17 June. On this latter day, larger differences are observed on the Net $F_{LW}$ compared to 16 June, which might be explained by the assumed profiles of gases such as $CO_2$, $O_3$ or water vapor, or the uncertainty in the LST. The $^{\uparrow}F_{LW}$ is highly dependent on the LST and, unfortunately, no LST data from MODIS were available on 17 June. A likely slight contamination by clouds on 17 June on the aircraft measurements may also partially explain this




larger difference on 17 June. On the other hand, the influence of the aerosol particles in the $F_{LW}$ is too weak to fully explain the observed differences, as the aerosol load was low (AOD at 550 nm ranging between 0.18 and 0.23).

[Figure 9]

A comparison of GAME results against the observations from ground-based pyrgeometer at Granada station and against CERES values obtained at the TOA is included in Figure 10. At the BOA, the diffuse radiation measured by the pyrgeometer is in quite good agreement with GAME calculations on 16 June, with differences within 1 $W·m^{-2}$. However, GAME overestimates the pyrgeometer data by 5 $W·m^{-2}$ (1.3%) on 17 June. These larger discrepancies might be related either to the likely cloud contamination, not accounted for in the model but affecting the pyrgeometer measurements, or to the selection of the LST values. At the TOA, GAME values are slightly larger than those observed by CERES. As for the SW, CERES closest data are around 2 hours and 600 km away from the measurement site, so only a qualitative comparison is possible. Due to the strong dependency of $^{\downarrow}F_{LW}$ to LST, the large variability in the satellite data and the difference in the horizontal coverage, differences of the order of 10 $W·m^{-2}$ observed on 16 June are considered reasonable. On 17 June, differences between GAME and the closest in time CERES values are of the order of 50 $W·m^{-2}$, as a result of the likely undetected cloud contamination in the satellite data.

[Figure 10]

As for the $ARE_{LW}$, Figure 11 shows the profiles obtained with GAME using the three datasets as inputs. Values at the BOA and TOA for each dataset and the average values are included in Table 7. Opposite to the SW, the $ARE_{LW}$ produces a heating effect both at the BOA and TOA, with positive values. The slight differences in the $F^{LW}$ in Figure 9 due to the use of different aerosol input datasets lead to variations of up to 2 $W·m^{-2}$ in the $^{BOA}ARE_{LW}$ (ranging from 20 to 26%), which needs to be considered in the interpretation of the results and reduced for a better estimate of the direct ARE. Despite this, values obtained for this dust event (3.2 $W·m^{-2}$ on average for both days) are in agreement with previous studies performed for mineral dust in the infrared region (Sicard et al., 2014a; 2014b; Meloni et al., 2018). It is extremely interesting to look at the differences between the two days in terms of AOD ($\Delta AOD$) and the effective radius for the coarse mode, $r_c$, ($\Delta r_c$) and their implication on the differences in the $ARE_{LW}$ at the



BOA ($\Delta^{BOA}ARE_{LW}$). For DS1 $\Delta AOD$ ($\Delta r_c$) is -0.02 (+0.18 μm) which produces a decrease in $^{BOA}ARE_{LW}$ ($\Delta^{BOA}ARE_{LW}$ = -0.5 W·m$^{-2}$). For DS2 $\Delta AOD$ ($\Delta r_c$) is -0.04 (+0.18 μm) which produces a decrease in $^{BOA}ARE_{LW}$ ($\Delta^{BOA}ARE_{LW}$ = -1.0 W·m$^{-2}$). If we relate these variations to the sensitivity study of Sicard et al. (2014a), in both cases the expected $ARE_{LW}$ increase due to the increase of the coarse mode radii is

counterbalanced by the $ARE_{LW}$ decrease when AOD decreases. Oppositely, for DS3 $\Delta AOD$ ($\Delta r_c$) is -0.05 (+0.64 μm), producing an increase of $^{BOA}ARE_{LW}$ ($\Delta^{BOA}ARE_{LW}$ = +1.6 W·m$^{-2}$). Here, the large increase of the coarse mode radius dominates over the AOD decrease. Sicard et al. (2014a) show indeed that the largest positive gradient of $ARE_{LW}$ occurs for median radii ranging from 0.1 to 2.0 μm. For DS3 the increase of $^{BOA}ARE_{LW}$ produced by a positive $\Delta r_c$ is larger than the decrease of $^{BOA}ARE_{LW}$ that would

have produced $\Delta AOD$ alone. At the TOA, same trends, but much less marked, are observed.

[Figure 11]

[Table 7]

## 4.3. Total mineral dust radiative effect

The total ARE, including both the SW and LW component, is included in Figure 12 and Table 8. As

observed, mineral dust produces a net cooling effect both at the surface and the TOA on both days. Depending on the input dataset used for the aerosol properties, values can change by up to 15 W·m$^{-2}$. On average, the $^{BOA}ARE$ values are -23.8 ± 8.4 and -29.2 ± 4.0 W·m$^{-2}$, and the $^{TOA}ARE$ is equal to -2.6 ± 2.2 and -7.0 ± 2.1 W·m$^{-2}$ on 16 and 17 June, respectively. These are 15 and 13% lower than for the SW spectral range, confirming that the LW fraction cannot be neglected. The $ARE_{LW}$ represents

approximately 20% of the $ARE_{SW}$ near the surface (except for DS3 on June 16), and reaches up to 50% at higher altitudes where the total ARE is quite low (see 16 June on Figure 12). The highest $ARE_{LW}/ARE_{SW}$ ratio at the TOA is obtained for the lowest values of $ARE_{SW}$ (DS2). Overall these $ARE_{LW}/ARE_{SW}$ ratios are in agreement with those found in previous studies for the Mediterranean region, which ranged between 9 and 26% (di Sarra et al. 2011; Perrone and Bergamo 2011; Sicard et al. 2014a).

[Figure 12]

[Table 8]

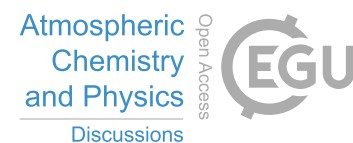

## 5 Conclusions

A moderate Saharan dust event affecting the western Mediterranean region during the Charmex/ADRIMED campaign on June 2013 was extensively monitored by ground-based and aircraft instrumentation above Granada experimental site. Radiative fluxes and mineral dust ARE both in the solar and infrared spectral ranges are calculated for this event with the RTM GAME. Three different aerosol input datasets, are used by GAME RTM in order to evaluate the impact of different input data in GAME calculations.

For the SW, very low variability with the input aerosol data (less than 1%) is observed for the radiative fluxes. The evaluation of GAME calculated radiative fluxes against the aircraft data reveals differences between the model fluxes and the measurements below 7%, with better agreement at altitudes above the planetary boundary layer. The differences between the retrievals with the three aerosol datasets are quite insignificant, especially taking into account the different approaches followed by the model and the pyranometers. Thus a conclusion on which input dataset provides a better performance is unlikely. The low discrepancies between GAME radiative fluxes retrievals lead to variations in the $ARE_{SW}$ of up to 33%, mostly driven by the differences in the aerosol vertical distribution and load, followed by the SSA.

For the LW, the effect of the aerosol in the radiative properties is lower compared to the SW, but certainly non-negligible and of opposite sign. GAME retrievals using the three aerosol datasets reveal differences in the fluxes lower than 2 W·m$^{-2}$ (less than 1%). The comparison with the pyrgeometer data measured at the ATR reveals however differences around 7%. Considering the low influence of the aerosol in the LW radiative fluxes, the influence of the assumed $CO_2$, $O_3$ and the used water vapor profiles and LST are needed to fully explain this discrepancy.

The total ARE, including both the SW and LW components, confirms that mineral dust produces a cooling effect both at the surface and the TOA, as already reported in the literature. On average, the $ARE_{LW}$ represents a 20% of the $ARE_{SW}$ at the surface, therefore clearly indicating that global model estimates need to consider the complete spectrum to avoid an overestimation on mineral dust cooling effect.



Additionally, it is necessary to be aware of the effects of using different measurement techniques and processing methodologies when calculating aerosol radiative properties. Even though the discrepancies observed here when using different aerosol datasets are slight, they still exist and a homogenization of the techniques to feed global models would be beneficial for a better estimate of the ARE and a reduced uncertainty.

## Acknowledgements

This work is part of the ChArMEx project supported by CNRS-INSU, ADEME, Météo-France and CEA in the framework of the multidisciplinary program MISTRALS (Mediterranean Integrated Studies aT Regional And Local Scales; http://mistrals-home.org/). Lidar measurements were supported by the ACTRIS (Aerosols, Clouds, and Trace Gases Research Infrastructure Network) Research Infrastructure Project funded by the European Union's Horizon 2020 research and innovation programme under grant agreement n. 654109. The Barcelona team acknowledges the Spanish Ministry of Economy and Competitivity (project TEC2015-63832-P) and EFRD (European Fund for Regional Development); the Department of Economy and Knowledge of the Catalan autonomous government (grant 2014 SGR 583) and the Unidad de Excelencia Maria de Maeztu (project MDM-2016-0600) financed by the Spanish Agencia Estatal de Investigación. The authors also thank the Spanish Ministry of Sciences, Innovation and Universities (ref. CGL2017-90884-REDT). This work was also supported by the Juan de la Cierva-Formación program (grant FJCI-2015-23904). P. Formenti and C. Denjean acknowledge the support of the French National Research Agency (ANR) through the ADRIMED program (contract ANR-11-BS56-0006). Airborne data was obtained using the aircraft managed by SAFIRE, the French facility for airborne research, an infrastructure of the French National Center for Scientific Research (CNRS), Météo-France and the French National Center for Space Studies (CNES). The authors acknowledge the use of GRASP inversion algorithm (www.grasp-open.com). The authors also kindly acknowledge Philippe Dubuisson (Laboratoire d'Optique Atmosphérique, Université de Lille, France) for the use of GAME model and Rosa Delia García Cabrera for her advice during the preparation of this manuscript.



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




**Tables and figures:**

|  | SW | LW |
|---|---|---|
| **Spectral range [µm]** | 0.297 – 3.100 | 4.5 - 40 |
| **Vertical range [km]** | 0-20 | 0-100 |
| **Number of levels** | 18 | 40 |
| **Vertical resolution (Vertical range) [km]** | 0.005 (0-0.01) | |
|  | 0.01 (0.01,0.05) | |
|  | 0.05 (0.05-0.1) | 1 (0-25) |
|  | 0.1 (0.1-0.2) | 2.5 (25-50) |
|  | 0.2 (0.2-1) | 5 (50-60) |
|  | 1 (1-2) | 20 (80-100) |
|  | 2 (2-10) | |
|  | 5 (10-20) | |

**Table 1. Summary of GAME main properties for the SW and LW spectral ranges. The altitude range corresponding to the different vertical resolution values is indicated between parentheses.**

| | | SW | | | LW | | |
|---|---|---|---|---|---|---|---|
| **Surface** | **alb** | AERONET | | | CERES | | |
| | **LST** | IISTA-CEAMA | | | MODIS | | |
| **Met. prof.** | **P,T,RH** | Aircraft + US std atm. | | | Aircraft + US std atm. | | |
| **Main gases** | **Conc.** | US std atm. | | | US std atm. | | |
| | **Abs.** | HITRAN | | | HITRAN | | |
| | | **DS 1** | **DS 2** | **DS 3** | **DS 1** | **DS 2** | **DS 3** |
| **Aerosol parameters** | **α$_{aer}$** | GRASP (z, 7 λ) | Klett (z, 3 λ) | Aircraft (z, 1 λ) | | Mie calculation | |
| | **SSA** | GRASP (z, 7 λ) | AERONET (col, 4 λ) | Aircraft (col, 1 λ) | | | |
| | **g** | AERONET (col, 4 λ) | AERONET (col, 4 λ) | AERONET (col, 4 λ) | | | |



**Table 2. Summary of the data sources used to obtain the input data parameterizations for GAME computations both in the SW and LW spectral ranges, including the surface parameters (albedo, *alb*, and Land-surface temperature, LST), profiles of meteorological variables and main gases and the aerosol parameters. For the aerosol parameters (aerosol extinction, $\alpha_{aer}$, single scattering albedo, SSA, and asymmetry parameter, g) three different datasets are used (DS1, DS2 and DS3) based on different instrumentation and retrievals. The indications below the sources of the aerosol parameters indicate whether the parameter is column integrated (col) or if it is vertically resolved (z) and the number of wavelengths at which it is given (n λ).**

|  | *alb*(440nm) | *alb*(675nm) | *alb*(870nm) | *alb*(1020nm) | *alb*(LW) | LST (K) |
|---|---|---|---|---|---|---|
| **June 16** | 0.05 | 0.15 | 0.30 | 0.30 | 0.016 | 314.5 |
| **June 17** | 0.05 | 0.15 | 0.31 | 0.31 | 0.013 | 298.1 |

**Table 3. Surface albedo, *alb(λ)*, values provided by AERONET for the SW spectral range and by CERES for the LW. Land-surface temperature (LST) on June 16 was obtained from MODIS whereas on June 17 was estimated from the meteorological station at Granada site. These surface parameters are common to all parameterizations.**

**16 June (SZA=31.49º)**

|  | $N_f$ | $N_c$ | $r_f$ | $r_c$ | $\sigma_f$ | $\sigma_c$ | AOD |
|---|---|---|---|---|---|---|---|
|  | (#·μm⁻²) | (#·μm⁻²) | (μm) | (μm) | (μm) | (μm) | (550 nm) |
| **DS1** | 9.04 | 0.018 | 0.12 | 2.22 | 0.48 | 0.73 | 0.18 |
| **DS2** | 7.53 | 0.014 | 0.12 | 1.90 | 0.57 | 0.65 | 0.23 |
| **DS3** | - | - | 0.11 | 1.92 | 0.63 | 0.66 | 0.23 |

**17 June (SZA=61.93º)**

|  | $N_f$ | $N_c$ | $r_f$ | $r_c$ | $\sigma_f$ | $\sigma_c$ | AOD |
|---|---|---|---|---|---|---|---|
|  | (#·μm⁻²) | (#·μm⁻²) | (μm) | (μm) | (μm) | (μm) | (550 nm) |
| **DS1** | 9.04 | 0.014 | 0.10 | 2.40 | 0.45 | 0.72 | 0.16 |
| **DS2** | 8.03 | 0.012 | 0.11 | 2.08 | 0.53 | 0.68 | 0.19 |
| **DS3** | - | - | 0.11 | 2.56 | 0.64 | 0.59 | 0.18 |

**Table 4. Column-integrated number concentration (*N*), effective radii (*r*) and standard deviation (σ) of fine and coarse aerosol modes and AOD at 550 nm for DS1, DS2 and DS3 on 16 and 17 June.**



| Mie calculations | | LW | | |
|---|---|---|---|---|
| | | **DS1** | **DS2** | **DS3** |
| | **RI** | DB (2017), (col, 601 λ) | DB (2017), (col, 601 λ) | DB (2017), (col, 601 λ) |
| | $r_{eff}$ | GRASP (col), | AERONET (col) | Aircraft (z) |
| | σ | GRASP (col) | AERONET (col) | Aircraft (z) |
| | $N$ | GRASP (z) | AERONET (col) | Aircraft (z) |

**Table 5. Summary of the data used to obtain $\alpha_{aer}(\lambda,z)$, $SSA(\lambda,z)$ and $g(\lambda,z)$ in the LW from Mie calculations, i.e. the refractive index, RI, effective radius, $r_{eff}$, geometric standard deviation, σ, and number concentration, $N$. Three different datasets are used (DS1, DS2 and DS3) based on different particle size distribution (PSD) data used. The indications below the sources of the aerosol parameters indicate whether the parameter is column integrated (col) or if it is vertically resolved (z) and the number of wavelengths at which it is given ($n$ λ). DB(2017) stands for Di Biagio et al., (2017).**

| | June 16 | | June 17 | |
|---|---|---|---|---|
| | $^{BOA}ARE_{SW}$ (W·m⁻²) | $^{TOA}ARE_{SW}$ (W·m⁻²) | $^{BOA}ARE_{SW}$ (W·m⁻²) | $^{TOA}ARE_{SW}$ (W·m⁻²) |
| **DS1** | -18.1 | -6.3 | -27.1 | -10.3 |
| **DS2** | -28.6 | -5.5 | -34.0 | -9.6 |
| **DS3** | -34.3 | -1.5 | -35.8 | -6.5 |
| **Avg ± std. dev** | -27.0± 8.2 | -4.4±2.6 | -32.3±4.6 | -8.8±2.0 |

**Table 6. ARE at the BOA and the TOA for the SW spectral range obtained with GAME using as inputs DS1, DS2 and DS3 for June 16 and 17, 2013. The averaged values and standard deviation are also included.**

| | June 16 | | June 17 | | ΔAOD | Δ$r_c$ (μm) | Δ$^{BOA}ARE_{LW}$ (W·m⁻²) |
|---|---|---|---|---|---|---|---|
| | $^{BOA}ARE_{LW}$ (W·m⁻²) | $^{TOA}ARE_{LW}$ (W·m⁻²) | $^{BOA}ARE_{LW}$ (W·m⁻²) | $^{TOA}ARE_{LW}$ (W·m⁻²) | | | |
| **DS1** | +3.1 | +2.2 | +2.6 | +1.6 | -0.02 | +0.18 | -0.5 |
| **DS2** | +3.9 | +2.9 | +2.9 | +1.7 | -0.04 | +0.18 | -1.0 |
| **DS3** | +2.5 | +1.3 | +4.1 | +1.8 | -0.05 | +0.64 | +1.6 |
| **Avg ± std. dev** | 3.2±0.7 | 2.1±0.8 | 3.2±0.8 | 1.7±0.1 | | | |




**Table 7. ARE at the BOA and the TOA for the LW spectral range obtained with GAME using as inputs DS1, DS2 and DS3 for June 16 and 17, 2013. The averaged values and standard deviation are also included. The last three columns include variations (Δ) of AOD, $r_c$ and ARE at the BOA between June 16 an 17 for the three datasets.**

|  | June 16 | | June 17 | |
|---|---|---|---|---|
|  | BOA ARE (W·m⁻²) | TOA ARE (W·m⁻²) | BOA ARE (W·m⁻²) | TOA ARE (W·m⁻²) |
| DS1 | -15.0 | -4.5 | -24.6 | -8.6 |
| DS2 | -24.7 | -3.1 | -31.1 | -7.8 |
| DS3 | -31.71 | -0.1 | -31.8 | -4.6 |
| Avg ± std. dev | -23.8±8.4 | -2.6±2.2 | -29.2±4.0 | -7.0±2.1 |

**Table 8. ARE at the BOA and the TOA for the total (SW+LW) spectral range obtained with GAME using as inputs DS1, DS2 and DS3 for June 16 and 17, 2013. The averaged values and standard deviation are also included.**

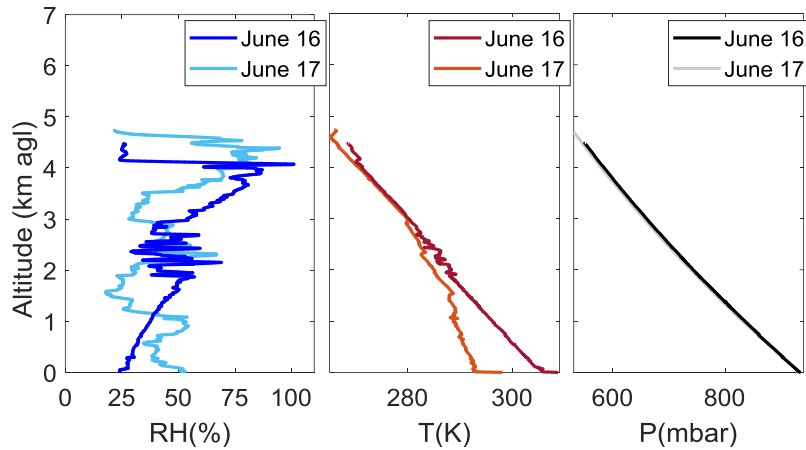

**Figure 1. Relative humidity (RH), temperature (T) and pressure (P) profiles measured on-board the ATR during flights F30 (June 16) and F31 (June 17).**





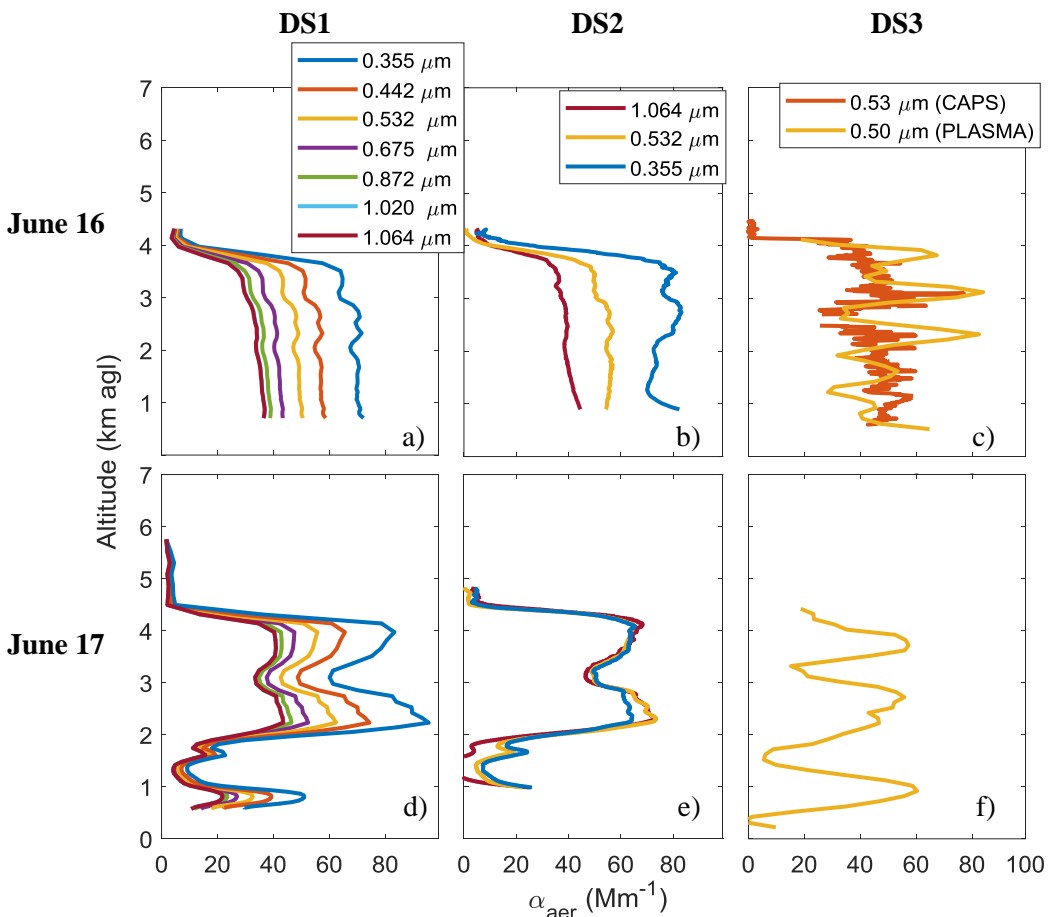

**Figure 2. Profiles of α$_{aer}$ obtained from GRASP/DS1 (left), Klett/DS2 (center) and aircraft in-situ/DS3 measurements (right) on June 16 (top row) and June 17 (bottom row).**

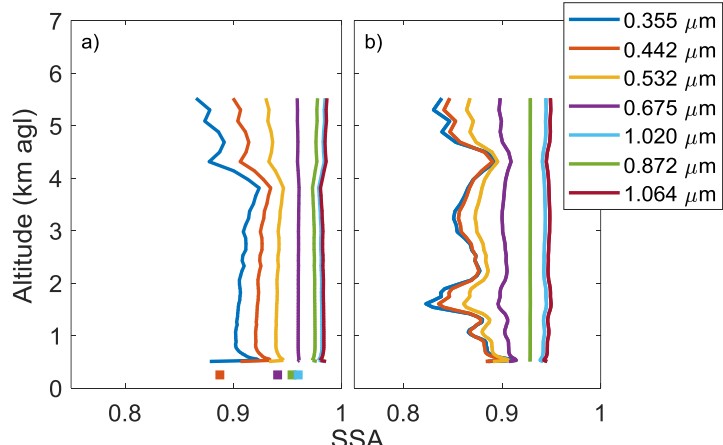



**Figure 3. SSA profiles obtained from GRASP/DS1 on June 16 (a) and June 17 (b).**

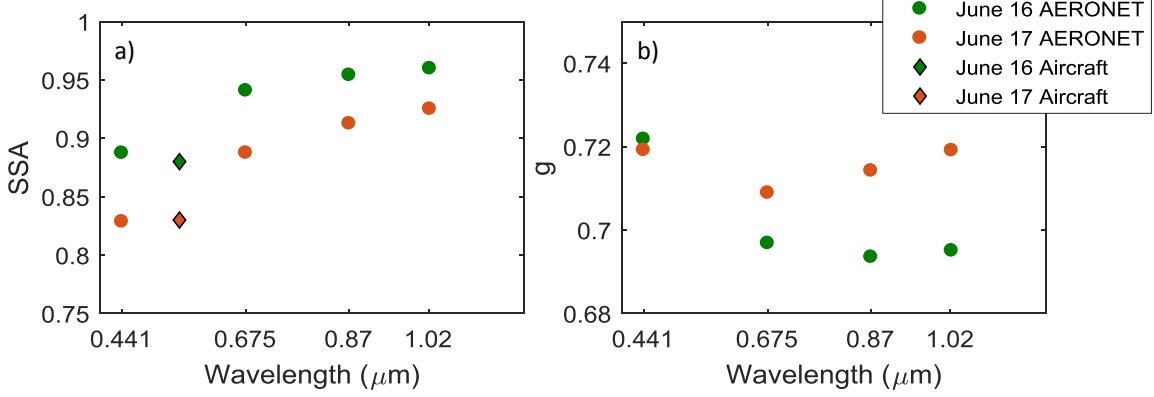

**Figure 4. a) AERONET/DS2 column-integrated (circles) and aircraft/DS3 averaged (diamonds) SSA values on June 16 at 16:22UTC and June 17 at 07:20UTC. b) AERONET g values for the same periods.**

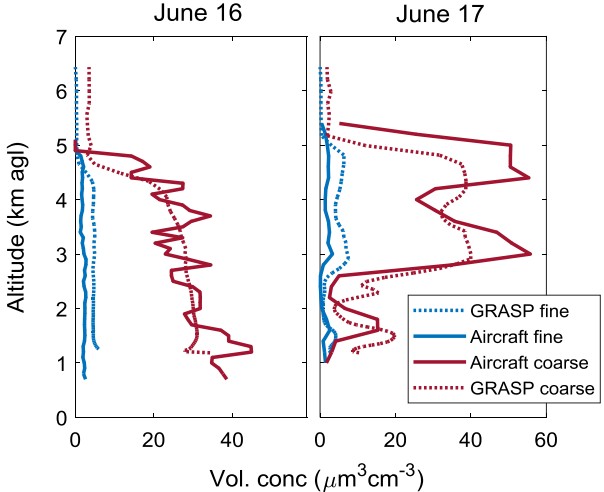

**Figure 5. Profiles of aerosol volume concentration for the fine (blue) and coarse (red) mode obtained from GRASP/DS1 (dotted line), and aircraft in-situ/DS3 measurements (solid line) on June 16 (left) and June 17 (right).**





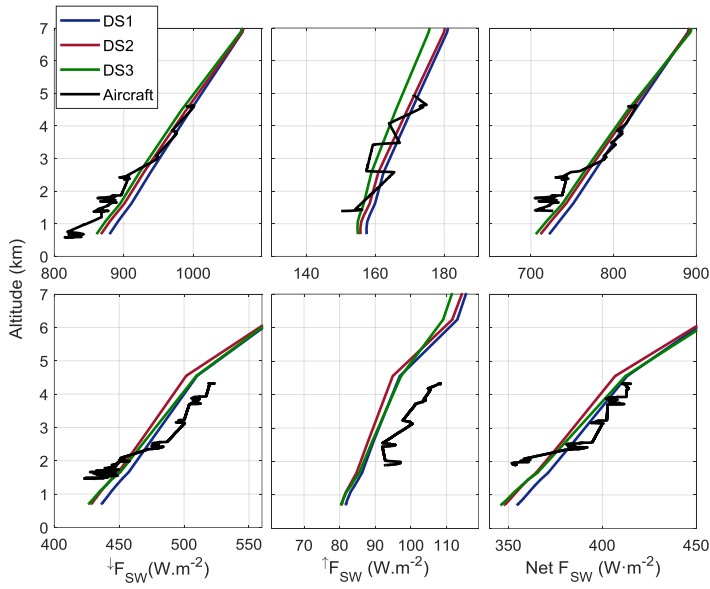

**Figure 6. Radiative fluxes for the SW spectral range for June 16 (upper row) and 17 (bottom row) simulated with GAME using different input aerosol datasets (DS1 in blue, DS2 in red and DS3 in green). The black lines are the aircraft in situ measurements**
5   **distant from about 20 km.**

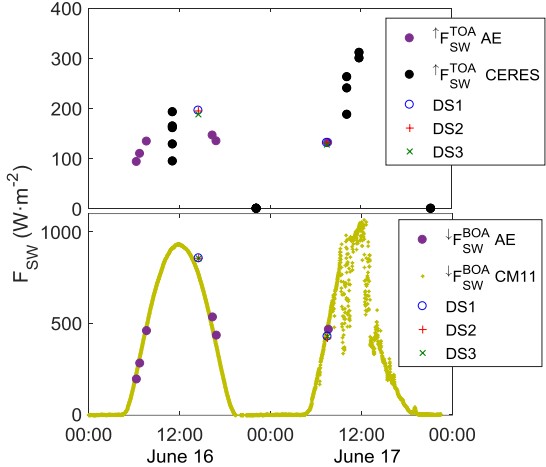

**Figure 7. Time series of the ↑F$_{SW}$ at the TOA (top) and ↓F$_{SW}$ at the BOA (bottom) for the period June 16-17. The green line represents surface measurements from the ground-based pyranometer at Granada station, purple dots are AERONET fluxes, black dots are CERES data and GAME ouput data for different inputs are represented by the blue circles (DS1), red (DS2) and green (DS3) crosses.**





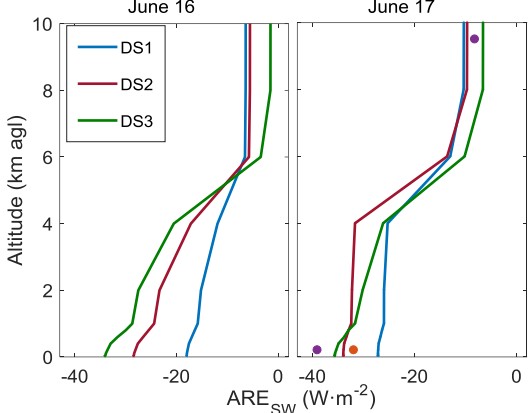

**Figure 8. ARE profiles in the SW spectral range simulated using DS1 (blue line), DS2 (red line) and DS3 (green line) as aerosol input data in GAME for June 16 (left) and June 17 (right). The purple dots represent the ARE provided by AERONET (AE) at the BOA and the TOA and the orange dot, the AERONET corrected for the surface albedo effect (AE-C; see text) ARE at the BOA.**

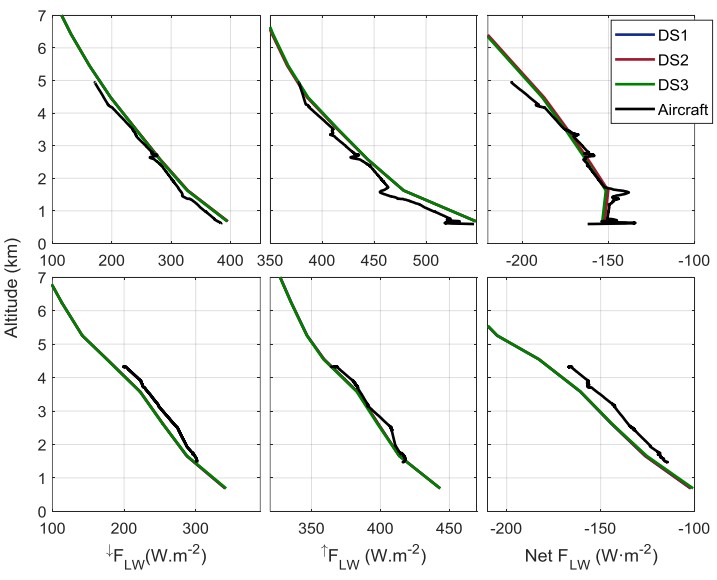

**Figure 9. Radiative fluxes for the LW spectral range for June 16 (upper row) and 17 (bottom row) simulated with GAME using different input aerosol datasets (DS1 in blue, DS2 in red and DS3 in green). The black line represents the aircraft in situ measurements.**





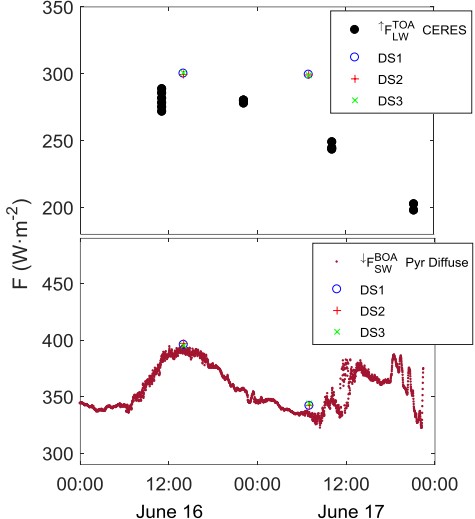

**Figure 10. Time series of the global $^\uparrow F_{LW}$ at the TOA (top) and the diffuse $^\downarrow F_{LW}$ at the BOA (bottom) during the period June 16-17. Surface measurements of diffuse (red) radiation from the ground-based pyranometer at Granada station are included. Black dots are CERES data and GAME ouput data for different inputs are represented by the blue circles (DS1), red (DS2) and green (DS3)**
5 **crosses.**

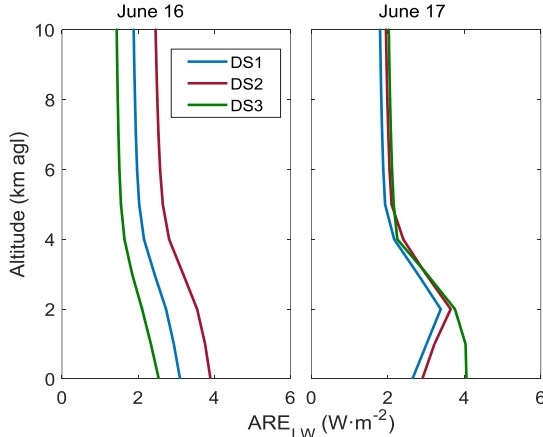

**Figure 11. Direct ARE profiles in the LW spectral range simulated using DS1 (blue line), DS2 (red line) and DS3 (green line) as aerosol input data in GAME for June 16 (left) and June 17 (right).**



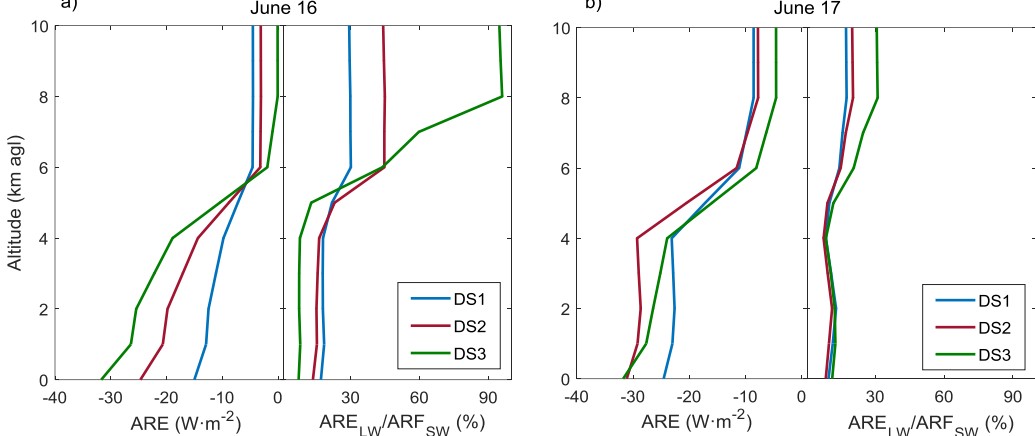

**Figure 12. Direct ARE for the total spectrum (left) and the ratio between the ARE LW and the ARE SW in percentage for DS1 (blue), DS2 (red) and DS3 (green) on June 16 at 14:30 UTC (a) and June 17 at 07:30 UTC (b)**

