# Peer review of "Impact of mineral dust on shortwave and longwave radiation: evaluation of different vertically-resolved parameterizations in 1-D radiative transfer computations"

_Atmospheric Chemistry and Physics, 2018_

## Referee Comment (RC1) · Anonymous Referee #1 · 30 Aug 2018

The paper presents an analysis of the optical and microphysical properties of dust particles observed from ground and from airplane on 16-17 June 2013 above southeastern Spain during the ChArMEx/ADRIMED campaign. The observations were conducted during a moderate Saharan dust event. Using a 1-D radiative transfer model, the author makes comparison of the output results obtained with different input data. They consider both shortwave and longwave radiation for the calculations. They concluded that the dust produces a cooling effect both at the surface and at the top of the atmosphere, as expected.

[Figure]

The paper in well written, the methodology and the results are clearly presented. The discrepancies coming from the different parametrizations are well analyzed. The authors conclude that global model estimate needs to consider the complete radiation spectrum to avoid an overestimation of the cooling effect produced by dust.

I have just one major concern. The same dust event was observed are almost the same location and at the same time by a balloon borne aerosol counter LOAC (Renard et al., Atmos. Chem. Phys., 18, 3677-3699, 2018, https://doi.org/10.5194/acp-18-3677-2018). Such counter measurements can be considered here for the estimate of the vertical distribution of the dust plume, and for the size distribution of the particles.

The paper can be published if the comments below are considered.

1. Abstract: A sentence must be added on the cooling effect found by the authors. 2. Instrument and data: Perhaps a map of the ground-based and airplane locations could be added. 3. Page 8 line 25: Such observation were also reported by Renard et al. 2018. 4. Page 10 line 9: The authors say that the concertation profiles of the main absorbing gas were taken from the US standard atmosphere. Nevertheless, real profiles can exhibit a significant variably from the standards for several reasons (local event, perturbed atmosphere…). Can you evaluate the effect of this variability on your results? 5. Page 10 line 15: The authors could consider the LOAC measurements, and the detection of large particles that produce a third mode. 6. Page 13 line 10: The author say that the refractive index of the dust are assumed to be constant with altitude. I understand that it is difficult to detect a possible variation of the index with altitude. Nevertheless, the authors must discuss the limit of this assumption, and how a variation of the refractive index can affect their results. They can consider the variability of the refractive index for different natures of dust and for the possible presence of pollution particles. 7. Page 13 line 12: Is it "interpolation" or "extrapolation"? 8. Page 13 line 23: The authors must also consider the LOAC balloon borne aerosol counting data. 9. Page 17 line 26: Do you think that the presence of large dust particles, not always detected from aircraft instruments, could partly explain the large differences you

observe?
* * *

---

## Referee Comment (RC2) · Anonymous Referee #2 · 5 Sep 2018

This work focuses on analyzing the differences in the aerosol radiative forcing obtained by the same radiative transfer model (GAME) using different datasets as input, during a dust intrusion in the Iberian Peninsula within the ADRIMED/CHARMEX campaign. The methodology that the authors use in this work is sound, and similar research has been already done in previous papers in order to analyze the sensitivities of radiative fluxes and aerosol radiative forcing (e.g. Gómez-Amo et al., 2010; 2011; Meloni et al. 2015; 2018), and aerosol heating rates (e.g. Meloni et al. 2015; Peris-Ferrús et al., 2017), to the aerosol properties used as input in the radiative transfer models. Despite this, the

most novel and interesting part of this work is the comparison between the results obtained using a very advanced and complete characterization of the aerosol properties, as well as their vertical distribution (GRASP) against those obtained from most known and widely used measurements and algorithms (i.e. Klett inversion lidar + photometer; and airborne in situ measurements). This reason is sufficient for this paper to be of interest for aerosol research in order to understand the uncertainties associated to aerosol radiative effect. Therefore, the argument of this paper is solid and then suitable for publication in ACP. However, I think there are several important aspects that can be improved, and it should be addressed before the paper is published.

GENERAL ASPECTS

In general, I miss a deeper analysis of the results, especially from a quantitative point of view. 1. Therefore, I would suggest that the authors focus their work on estimating the sensitivity of the GAME model to the different aerosol inputs., answering the following question that underlies their own figures: why the authors observe notable differences in the ARE among the datasets, in shortwave and longwave ranges, despite the differences in the vertical profiles of radiative fluxes they obtain are negligibles?. This should be done in a quantitative way by taken into account the differences among the aerosol properties used in the three datasets.

2. For this, I think that the differences among the aerosol datasets used should be better explained, in terms of the aerosol properties (i.e. extinction, absorption and scattering):

If I understand well the inputs that GAME model requires for aerosol characterization ext(wavelength,z), SSA(wavelength,z) and asymmetry parameter(wavelength,z): a. In the shortwave range DS1 - GRASP provides the spectral profiles (7 wavelengths) of the aerosol extinction and SSA. DS2 - However, the Klett inversion only provides the spectral (3 wavelengths) extinction profile (taking into account vertically constant LR). The SSA is constant with height and column-integrated AERONET values (4 wave-

lengths) are assumed. DS3 - Airborne measurements also provide information about extinction and absorption profiles; with no spectral considerations. In the three cases the column-integrated AERONET asymmetry parameter (4 wavelengths) is assumed. This information is well summarized in Table 2, but I miss better explanation in the text.

On the other hand, there is a different aerosol layering among the studied days that can play an important role on the retrieved ARE. Looking at the aerosol extinction profile (Figure 2) and the concentration of Fine and Coarse modes (Fig. 5): June 16, a single homogeneous aerosol layer is observed June 17, aerosol are uncoupled in two layers The same is observed in the SSA profiles shown in Fig. 3. have you consider to analize the role of aerosol layering in your retrieval?

b. In the longwave range. The authors obtain the aerosol properties by Mie calculations as appears in Tables 2 and 4. However, it is not clear what radius are used in Mie calculations. Sometimes the authors assert they use the reff and nevertheless, in table 4 the radii appear in the 2 modes (fine and coarse). Please be clear and consistent.

3. The results should be analized taking into account the quantitative differences among the aerosol properties used in the aerosol datasets, considering - spectral variation - vertical variation

Considering that the main differences among the aerosol datasets are based on differences in the vertical profile of extinction and absorption, the authors should take into account the work already published in this regard, using other models and different datasets. For example, to help in the interpretation of the differences observed in the shortwave, I would recommend reading of: Meloni et al. 2005. Where the effect of the extinction profile on the calculation of the ARE is analyzed. Different works by Gómez-Amo et al., 2010 and 2011, as well as by Guan et al., 2010. Where the effect of vertically varying the aerosol absorption in the determination of ARE is analyzed.

Main concerns about results and conclusions sections: SW: Is difficult to understand that with such small differences among the different dataset input (below 1% for radiative fluxes and 0.05 for AOD), why do the authors obtain such large differences in the AREsw (up to 33%)? I think that this is the question you have to answer in this paper, using your data and simulations, which is missing in this paper. At fixed solar zenith angle, the shortwave fluxes are mainly dependent on the AOD. The direct flux is totally driven by the extinction (AOD) and with such small AOD variations between datasets I do not expect large differences in the fluxes (just what you obtain and is shown in Fig. 5 and 6). However, the diffuse radiation is extremely dependent on SSA and the phase function (i.e. asymmetry parameter). If AOD and asymmetry parameter remain fixed, Gómez-Amo et al., (2010) showed that the differences observed in the ARF (at the surface and TOA) are driven by the vertical distribution of SSA that results in different distribution of the diffuse radiation. I would suggest repeating the analysis by removing the small variation of AOD. For example by normalizing the three datasets to the AERONET AOD, or working with the forcing efficiency, and focus the analysis on the variations due to the SSA.

I think it would be useful for the interpretation of the results: - to show in Fig. 4 the spectral variation of SSA for the 2 layers observed on June 17, and for the homogenous layer on June 16. - the vertical profiles of SW and LW fluxes in aerosol-free conditions should be shown in Fig. 6 and 9., respectively. (see Meloni et al., 2015; 2018)

LW: P20-L20: I do not understand this sentence: "Considering the low influence of the aerosol in the LW radiative fluxes, the influence of the assumed $CO_2$, $O_3$ and the used water vapor profiles and LST are needed to fully explain this discrepancy". Why do you think that the differences in the LW fluxes are due to the assumed $CO_2$, $O_3$ and the used water vapor profiles and LST? Did you change them among the simulations DS1, DS2 and DS3? According to table 4, these values do not change with the dataset considered

Fig 12. The authors report an ARE offset LW/SW increase with altitude, up 90% at higher altitudes, when there was not aerosol layer anymore. This is totally opposite at what is reported in Meloni et al., (2015), that found the maximum offset at the surface

and a negligible variation from the top of the aerosol layer to the TOA. These results should be better discussed and justified.

Minor comments:

P2-L3: Please change "contrasted" by "compared" P2-L21: Please rephrase the sentence, its meaning is no clear. P4-L5: Please change "..model estimates sensitivity..." by "..sensitivity of the model estimates.." P5-L2: Please change "..real and imaginary refractive indices..." by "..real and imaginary parts of the refractive index.." P5-L24: Plase remove "particle" P5-L29: Please change "..spatial integral..." by "..vertical integration.." P6-L6: Please change "in" by "by" P9-L27: How the surface temperature was estimated from the measurements at 2m above the ground? please provide a reference. P11-L12: This sentence is really surprising. I do not understand well the differences in the AOD among the datasets reported in table 4. Since the AOD measured with the CIMEL photometer is imposed as a closure condition in the GRASP and Klett inversions, one would expect similar AOD for DS1 and DS2 datasets, contrarily to what is reported in Table 4. On the other hand, AOD differences are expected from DS1 and DS2, with respect to DS3 (aircraft exctinction). These differences may be also due to the AOD content from surface to the minimun altitude of the aircraft, or for the observation of different airmasess (20km far from ground-based station and aircraft). I think the authors should clearly discuss these AOD differences, since the discussion about the differences in the ARE results is mainly addressed in terms of AOD. P13-L2: This approach is similar to the used in Meloni et al., (2015,2018) and Peris-Ferrús et al., (2017). This papers should be cited in this paragraph. P13-L11: Are you sure about this sentence? Is there any typo error? wavelengths below 3 um are not considered LW range. What about for wavelengths over 16 um? P13-L14: What radius did you use in Mie calculations? the effective radius, or the rF and rC? Table 4, Table 4 caption and the text result confusing. Please be clear. P14-L11. Please change "visible" by "shortwave" P14-L12. Please change "thermal" by "longwave" P14-L27. When the authors talk about "discrepancies", are they refeering to "relative differences" (Fgame

- Faircraft)/Faircraft? The authors should define how they obtain these discrepances, and always use the same term. Sometimes they use "discrepancies" and sometimes "differences". P15-L5. Please clarify the meaning of this last sentence of the paragraph. The main aerosol effect over the radiative fluxes is due to the AOD, the SSA is a sencond order effect. P15-L11. I do not see the influence of the boundary layer due to the distance between the ground-based station and the aircraft leg. The DS3 simulation is set with the aircraft data, so the 20 km distance between the ground-based station and the aircraft leg should be reflected in the results obtained from the DS3 simulation. P15-L17. The authors should take into account that relative differences about 7% for a Fdownward around 430 and 530 Wmˆ-2 yield absolute differences of 30 and 37 Wmˆ-2. These differences may represent a large fraction of the aerosol effect, with a large contribution to the uncertainties in the determination of ARE using this data. I don't think that these differences are quite insignificat as the authors assert. Have you evaluated them? P15-L27. Again, a 60 Wmˆ-2 differences between datasets may contribute to large uncertainties in the determination of ARE using this data. Please evaluate this. P16-L14. I think that the comparison GAME-CERES have no sense, even qualitatively, with CERES overpass 600km. The upward flux at TOA is mainly dependent on surface albedo and clouds, and both can be really different at 600km away. I would suggest remove this comparisons, since they do not present a relevant contribution to this work. P17-L8:L15. Since this paper is mostly to analyze differences in ARE using different datasets. The authors should carry out a more in deep analysis of the results. Specially regarding to the surface and TOA values. Eeven if the values obtained for the three datasets fall within the range of the values previosuly observed in the Mediterranean, notice that the values shown in Table 6 differ from a 30-50%. These differences are really large and dependent on the used dataset, and consequently it should be analyzed in detail. Please take into account the following references to help with the interpretation. P17-L21. Please change "..from Mie calculations from..." by "..from Mie calculations for.." P17-L23:P18-L4. I think that a more detailed analysis is needed to establish why you observe these differences between GAME and aircraft

measurements. On 16 June, the differences may be explained "by the assumed profiles of gases such as $CO_2$, $O_3$ or water vapor, or the uncertainty in the LST", as the authors assert, or may be not. It is just a too simple qualitative explanation. P18-L7. Please change "the diffuse radiation..." by " the longwave radiation.." P18-L9. The differences observed are within the uncertainties of the pirgeometers, that for a well mantained and calibrated instrument are below 5 Wmˆ-2 (Meloni et al., 2012).Therefore, the authors does not need qualitative explanations based on variables as LST to justify the differences. P18-L11. As in the SW case, I think it is not worth including the comparison with CERES due to the large distance between the ground-based and satellite measurements. P18-L28. Please be consitent, rc or reffc?

Tables 6, 7, 8: The standard deviation is an statistical parameter then it have not sense if obtained over three values only.

REFERENCES: - Meloni, D., A. di Sarra, T. Di Iorio, G. Fiocco, Influence of the vertical profile of Saharan dust on the visible direct radiative forcing, Journal of Quantitative Spectroscopy & Radiative Transfer 93 (2005) 397–413 - Meloni, D., Di Biagio, C., di Sarra, A., Monteleone, F., Pace, G., and Sferlazzo, D. M.: Accounting for the solar radiation influence on downward longwave irradiance measurements by pyrgeometers, J. Atmos. Ocean. Technol., 29, 1629–1643, 2012. - Gómez-Amo, J.L., A.diSarra, D.Meloni, M.Cacciani, M.P.Utrillas, Sensitivity of shortwave radiative fluxes to the vertical distribution of aerosol single scattering albedo in the presence of a desert dust-layer, Atmospheric Environment 44 (2010) 2787-2791. - Gómez-Amo, J.L., V. Pinti, T. Di Iorio, A. di Sarra, D. Meloni, S. Becagli, V. Bellantone, M. Cacciani, D. Fuà, M.R. Perrone, The June 2007 Saharan dust event in the central Mediterranean: Observations and radiative effects in marine, urban, and sub-urban environments, Atmospheric Environment 45 (2011) 5385-5393 - Peris-Ferrús, C., J.L. Gomez-Amo, C. Marcos, M.D. Freile-Aranda, M.P. Utrillas, J.A. Martínez-Lozano, Heating rate profiles and radiative forcing due to a dust storm in the Western Mediterranean using satellite observations, Atmospheric Environment 160 (2017) 142-153. - Guan, H., B. Schmid, A. Bucholtz,

and R. Bergstrom, Sensitivity of shortwave radiative flux density, forcing, and heating rate to the aerosol vertical profile, JOURNAL OF GEOPHYSICAL RESEARCH, VOL. 115, D06209, doi:10.1029/2009JD012907, 2010

---

## Referee Comment (RC3) · Anonymous Referee #3 · 14 Sep 2018

The paper examines a case of a moderate dust intrusion in Granada with the aim of calculating the dust radiative effect either in the SW and in the LW regions at the surface, at the top of the atmosphere, and within the dust layer by means of the GAME radiative transfer model. The focus of the study is the sensitivity of the SW and LW radiative fluxes and effects on the dust microphysical and optical properties derived in three different ways, using a combination of remote sensing (AERONET and lidar) measurements and the GRASP inversion code, and in situ airborne measurements from the SAFIRE ATR42 aircraft. This study is carried out in the framework of the

[Figure]

ChArMEx/ADRIMED campaign and takes advantage of the large observation efforts placed during the project, using either ground-based, airborne and satellite observations. The results of the model calculations are those expected (SW cooling and LW heating by dust, with a not negligible LW/SW ratio), and the case study is not that of an extraordinary dust transport (AOD moderately low). However, the most important conclusion provided by the authors is that optical properties derived from different measurements and with different techniques may provide non-negligible differences in the radiative effects, and should be of concern when estimating dust forcing. I recommend publication, but after some major issues are resolved by the authors.

Major issues The main issue on the presented results concerns the simulation of the SW irradiances using the three different aerosol optical properties DS1, DS2, and DS3, and the evaluation of the ARE. The authors found that very close SW irradiances at surface are obtained with the three datasets (the values seem coincident in Figure 7, but no quantitative information is provided in the text), while differences in the vertical profiles of the downward SW irradiances simulated with GAME are visible in Figure 6 close to the surface when using DS1 or DS2 (which seems to provide identical irradiances than DS3). So there seems to be an inconsistency between the simulations of the vertical profiles and of the surface SW irradiance. As a second point, the LW ARE is very low, due to the values of the AOD, and they might be comparable to the model uncertainties. So a careful evaluation of the uncertainties on the model output should be performed. The fact that the net LW irradiance on 17 June is overestimated by the model is likely not a problem of the $CO_2$ and $O_3$ profiles (the water vapor one should have been taken into account in the simulation, as it measured during the flights) used in the model (to my knowledge the impact of this minor gas is negligible) , but comes from the fact that NET=FâĘŞ-FâĘŚ, and FâĘŞ is overestimated, while FâĘŚ is comparable to measurements. The model overestimation of the LW irradiance at the surface on 17 June (which, however, is as large as the measurement uncertainty) cannot be due to clouds affecting the measurements and not accounted in the model, because clouds increase the LW irradiance, and this should cause a model underestimation. I

suggest to perform a sensitivity study to assess the uncertainty on the modelled SW and LW irradiances due to the uncertainty of the input parameters. As the authors state, the CERES observations are too far in space (600 km) and in time (2 hours) to the surface observations, and a quantitative comparison with the RT model simulations is not possible. I think that the approximate results on the TOA fluxes using CERES data should be removed.

Minor issues Introduction: a quantitative description of the SW and LW dust radiative effect in the Mediterranean from previous studies is missing. Page 6, line 9: remove "diffuse" before downward radiative fluxes for the LW. Page 7, line 10: the authors should better explain how measurements corresponding to large pitch and roll aircraft angles are filtered. Moreover, this should be applied not only to downward pyrgeometer measurements, but to either pyranometers and pyrgeometers, both downward and upward-looking. The description of the correction of the ATR42 radiation measurements for the variation in the solar position and in the aircraft attitude during flight applies to pyranometers and not the pyrgeometer measurements, as stated in line 10. The final uncertainty in the airborne SW irradiance profiles is not reported. Page 8, lines 23-24: a figure showing the airmass back trajectories may be useful. Page 8, line 26: change "profile" with "flight". Do the same at Page 9, line 5. Page 9, lines 26-27: the authors mean that, due to the lack in MODIS data of LST the air temperature 2 m a.g.l. have been used as surrogate of the LST? This may lead to a relevant underestimation of the upward LW irradiance. Did the authors verify the differences in air and surface temperature in other cases and the impact of using one or the other in the simulation of the upward LW irradiances? Page 13 lines 11-12: maybe the authors mean "extrapolation" instead of "interpolation" and "longer" instead of "shorter". Page 14 line 11: use "solar" and not "visible". Page 15 line 2: the sentence "Even though the discrepancies in the AOD are within the uncertainty..." is misleading. DS1 and DS2 use AERONET AOD measurements as input, so both should agree within the measurement uncertainty ($\pm$0.01) at 550 nm. But a difference of 0.05 is reported in Table 4. This difference should be addressed and explained. Page 15 lines 5-7:

the spectral variation of the SSA is provided only for the wavelengths at which it has been derived (Figure 4). For example DS3 from aircraft measurements is provided only for one wavelength. This not helps in understanding how optical properties may influence the SW radiation in the whole interval. Page 15 lines 8-9: this sentence is rather obvious, since the comparison with aircraft radiation fluxes can be done only for the altitudes covered by the ATR42. Page 15 line 13: maybe the authors mean "below" instead of "above". Page 15 lines 25-28: the authors should present and discuss absolute and relative differences between measurements, AERONET calculations and GAME simulations. I don't understand the sentence stating that on 16 June the radiation presents large values (than 17 June?), please explain. An overestimation by 6% by GAME is much larger than the irradiance uncertainty. The estimation of the simulated irradiances is necessary in order to understand if such a difference is within the model and measurement respective uncertainties, otherwise it should be commented and possible causes addressed. Page 17 lines 8-15: the SZA at which the RTM simulations are performed should be cleared. Are they those corresponding to the middle of the F30 and F31 flights (31.49° for 16 June and 61.93° for 17 June?). The SW ARE depends on the SZA, so when the authors presents the ARE from previous studies are they sure that such values can be direct compared with theirs? I don't think so, since results from Papadimas et al. (2012) are regional summer means. The same goes for the comparison with the other references (Sicard et al, 2014a,b; Barragan et al., 2017). Moreover, ARE depends on AOD. Thus, when comparing cases with different AOD, the radiative forcing efficiency (ARE per unit AOD) may be the most appropriate quantity. Page 18, line 7: change "diffuse" with "longwave". Page 18, line 13: the upward and not the downward component of the LW irradiance depends on LST. Page 30, Table 3: albedo in not defined in the LW region! Do authors refer to emissivity? Is it spectrally integrated or for a single wavelength? Page 34, Figure 5: a shift is visible in the volume concentration profile of 17 June form GRASP code and aircraft measurements. Can the authors comment on it? Page 35, Figure 6: The uncertainty range of the radiative fluxes from airborne instrumentation should be added in the plot to help understanding

how much model calculations agree with measurements. The same for Figure 9.

---

## Author Comment (AC1) · 7 Nov 2018

We would like to thank the reviewers for their efforts and thorough review of our manuscript. We realize that the notes and suggestions made will improve the quality of the paper. Hereafter, the reviewers' comments are presented in bold font and the text included in the manuscript is marked in italics. Line numbering is referred to the new version of the manuscript.

**The paper presents an analysis of the optical and microphysical properties of dust particles observed from ground and from airplane on 16-17 June 2013 above southeastern Spain during the ChArMEx/ADRIMED campaign. The observations were conducted during a moderate Saharan dust event. Using a 1-D radiative transfer model, the author makes comparison of the output results obtained with different input data. They consider both shortwave and longwave radiation for the calculations. They concluded that the dust produces a cooling effect both at the surface and at the top of the atmosphere, as expected.**
**The paper in well written, the methodology and the results are clearly presented. The discrepancies coming from the different parametrizations are well analyzed. The authors conclude that global model estimate needs to consider the complete radiation spectrum to avoid an overestimation of the cooling effect produced by dust.**
**I have just one major concern. The same dust event was observed are almost the same location and at the same time by a balloon borne aerosol counter LOAC (Renard et al., Atmos. Chem. Phys., 18, 3677-3699, 2018, https://doi.org/10.5194/acp-18-3677-2018). Such counter measurements can be considered here for the estimate of the vertical distribution of the dust plume, and for the size distribution of the particles.**

The ballon borne measurements included in Renard et al., (2018) were performed at the stations of Minorca and Ile du Levant. These locations are around 700 km away from Granada, therefore, no data are available at Granada station where the current study is performed.

**The paper can be published if the comments below are considered.**
   1. **Abstract: A sentence must be added on the cooling effect found by the authors.**

Done.
Page 2, lines 6-7: *"The three parameterization datasets produce a cooling effect due to mineral dust both at the surface and the top of the atmosphere."*

**2. Instrument and data: Perhaps a map of the ground-based and airplane locations could be added.**

The map is already included in Benavent-Oltra et al., 2017 (Figure 1), as indicated in the manuscript (Page 6, line 24).

**3. Page 8 line 25: Such observation were also reported by Renard et al. 2018.**

The study by Renard et al. (2018) is included as a reference in the manuscript where appropriate.

**4. Page 10 line 9: The authors say that the concertation profiles of the main absorbing gas were taken from the US standard atmosphere. Nevertheless, real profiles can exhibit a significant variably from the standards for several reasons (local event, perturbed atmosphere...). Can you evaluate the effect of this variability on your results?**

Variations in the concentration of the main absorbing gases have a low impact on the radiative fluxes profiles and much lower impact in the ARF, since only the effect of the aerosol is accounted for. As an example, we performed a sensitivity test varying the ozone profiles up to double concentrations. Comparing the obtained results with those obtained for DS1 using the standard concentration, we observe differences lower than 4 $W \cdot m^{-2}$ in the $^{\downarrow}F_{SW}$ and lower than 2 $W \cdot m^{-2}$ in the case of the $^{\uparrow}F_{SW}$. For the LW, differences lower than 1.5 $W \cdot m^{-2}$ in the $^{\downarrow}F_{LW}$ and lower than 3.6 $W \cdot m^{-2}$ in the case of the $^{\uparrow}F_{LW}$ are obtained. For the ARF differences are negligible (below 0.2 $W \cdot m^{-2}$).

[Figure]

**Figure. Radiative fluxes for the SW (top) and LW (bottom) spectral range for June 16 simulated with GAME using DS1 (in blue) and DS1 with double ozone concentration (orange dashed line). The black lines are the aircraft in situ measurements.**

[Figure]

**Figure. ARE profiles in the SW (right) and LW (left) spectral ranges simulated using DS1 (blue line) and DS1 with double ozone concentration (orange dashed line) as input data in GAME for June 16.**

**5. Page 10 line 15: The authors could consider the LOAC measurements, and the detection of large particles that produce a third mode.**

We agree with the reviewer that the LOAC would provide very valuable information for our study. Nonetheless, no LOAC measurements are available above Granada site.

**6. Page 13 line 10:**
**The author say that the refractive index of the dust are assumed to be constant with altitude. I understand that it is difficult to detect a possible variation of the index with altitude. Nevertheless, the authors must discuss the limit of this assumption, and how a variation of the refractive index can affect their results. They can consider the variability of the refractive index for different natures of dust and for the possible presence of pollution particles.**

For our study we assumed the refractive index provided by Di Biagio et al., (2017) for the Moroccan source, which we consider as the most appropriate for our case studies. By using the refractive index provided for the Algerian and Mauritanian sources from the same database we observe variations in the ARF at the BOA of 0.8 $W \cdot m^{-2}$ and 0.3 $W \cdot m^{-2}$ at the TOA. These results come to confirm the importance of having an accurate aerosol database for the retrieval of the ARF and the reduction of the uncertainties. They have been reported in the new version of the manuscript:
Page 14, lines 20-25: *"This assumption is not exempt of uncertainty, since the refractive index present a certain variability associated to the different nature of mineral dust properties. For example, the use of the refractive index provided for the Algerian and Mauritanian sources from Di Biagio et al., (2017) leads to variations in the ARF of 0.8 and 0.3 $W \cdot m^{-2}$ at the BOA and the TOA respectively. Additionally, vertical variations of the refractive index are also a source of uncertainty in the obtained radiative fluxes."*

**7. Page 13 line 12: Is it "interpolation" or "extrapolation"?**

Text has been modified. Interpolation has been replaced by extrapolation.

**8. Page 13 line 23: The authors must also consider the LOAC balloon borne aerosol counting data.**

See our previous responses on this topic.

**9. Page 17 line 26: Do you think that the presence of large dust particles, not always detected from aircraft instruments, could partly explain the large differences you observe?**

We agree that the presence of large particles may partly explain the differences observed in the longwave radiative fluxes. As stated in the introduction, large particles are especially relevant for the aerosol radiative forcing in the longwave range. Therefore, it is important to consider the presence of these large particles in the aerosol measurements required to feed radiative transfer models. As an example, we performed a simulation with our model considering a third mode with similar characteristics to the one observed during this dust event in Minorca. On June 16, by adding this third mode (assuming that the number concentration is a 15% of the coarse mode) we observe an increase in the $ARF_{LW}$ of $0.5 W \cdot m^{-2}$ at the BOA and $0.25 W \cdot m^{-2}$ at the TOA. This increase in the ARE due to the presence of a third mode is significant, but it does not fully explain the differences observed here.

---

## Author Comment (AC2) · 7 Nov 2018

We would like to thank the reviewers for their efforts and thorough review of our manuscript. We realize that the notes and suggestions made will improve the quality of the paper. Hereafter, the reviewers' comments are presented in bold font and the text included in the manuscript is marked in italics. Line numbering is referred to the new version of the manuscript.

**This work focuses on analyzing the differences in the aerosol radiative forcing obtained by the same radiative transfer model (GAME) using different datasets as input, during a dust intrusion in the Iberian Peninsula within the ADRIMED/CHARMEX campaign. The methodology that the authors use in this work is sound, and similar research has been already done in previous papers in order to analyze the sensitivities of radiative fluxes and aerosol radiative forcing (e.g. Gómez-Amo et al., 2010; 2011; Meloni et al. 2015; 2018), and aerosol heating rates (e.g. Meloni et al. 2015; Peris-Ferrús et al., 2017), to the aerosol properties used as input in the radiative transfer models. Despite this, the most novel and interesting part of this work is the comparison between the results obtained using a very advanced and complete characterization of the aerosol properties, as well as their vertical distribution (GRASP) against those obtained from most known and widely used measurements and algorithms (i.e. Klett inversion lidar + photometer; and airborne in situ measurements). This reason is sufficient for this paper to be of interest for aerosol research in order to understand the uncertainties associated to aerosol radiative effect. Therefore, the argument of this paper is solid and then suitable for publication in ACP. However, I think there are several important aspects that can be improved, and it should be addressed before the paper is published.**

**GENERAL ASPECTS**
**In general, I miss a deeper analysis of the results, especially from a quantitative point of view.**
**1. Therefore, I would suggest that the authors focus their work on estimating the sensitivity of the GAME model to the different aerosol inputs., answering the following question that underlies their own figures: why the authors observe notable differences in the ARE among the datasets, in shortwave and longwave ranges, despite the differences in the vertical profiles of radiative fluxes they obtain are negligibles?. This should be done in a quantitative way by taken into account the differences among the aerosol properties used in the three datasets.**
**2. For this, I think that the differences among the aerosol datasets used should be better explained, in terms of the aerosol properties (i.e. extinction, absorption and scattering): If I understand well the inputs that GAME model requires for aerosol characterization ext(wavelength,z), SSA(wavelength,z) and asymmetry parameter(wavelength,z):**
**a. In the shortwave range DS1 - GRASP provides the spectral profiles (7 wavelengths) of the aerosol extinction and SSA. DS2 - However, the Klett inversion only provides the spectral (3 wavelengths) extinction profile (taking into account vertically constant LR). The SSA is constant with height and column-integrated**

**AERONET values (4 wave-lengths) are assumed. DS3 - Airborne measurements also provide information about extinction and absorption profiles; with no spectral considerations. In the three cases the column-integrated AERONET asymmetry parameter (4 wavelengths) is assumed. This information is well summarized in Table 2, but I miss better explanation in the text.**

The differences in ARE observed between the three datasets are comparable to those observed in the radiative fluxes in absolute terms and therefore the results are consistent. A thorough quantitative discussion on the differences between the aerosol properties was already presented in Benavent-Oltra et al. (2017) and thus it was out of the scope of the present study. However, additional information has been added to the manuscript for completeness considering the reviewer's suggestion. Additionally, a small paragraph summarizing Table 2 and Section 3.2.2 has been added.

Page 14, lines 1-8: *"Summing up, for the SW aerosol parametrization in GAME three datasets are tested. In DS1, GRASP provided spectral profiles at 7 wavelengths of the aerosol extinction and SSA are used. In DS2, the Klett retrieved extinction profiles at 3 wavelengths are used together with the AERONET SSA columnar values at 4 wavelengths, which are assumed to be constant with height. For DS3, one extinction profile at 550 nm and a column-averaged single-wavelength value of the SSA from the airborne measurements are considered. In the three cases, the column-integrated AERONET asymmetry parameter at 4 wavelengths is assumed to be constant with height and used as input."*

**On the other hand, there is a different aerosol layering among the studied days that can play an important role on the retrieved ARE. Looking at the aerosol extinction profile (Figure 2) and the concentration of Fine and Coarse modes (Fig. 5): June 16, a single homogeneous aerosol layer is observed June 17, aerosol are uncoupled in two layers. The same is observed in the SSA profiles shown in Fig. 3. have you consider to analize the role of aerosol layering in your retrieval?**

We thank the reviewer for this suggestion. Because of the different SZA during both days and the different atmospheric conditions, a direct comparison between June 16 and 17 in order to study the influence of the aerosol layering is not possible. However, additional comments on the aerosol layering influence on the ARE are now included in the text considering the information available in the literature (e.g. Meloni et al., 2004; Guan et al., 2010). These studies show that the vertical structure of the aerosol has lower impact in the ARF at the surface level for low-to-moderate absorbing particles, which should be our case for mineral dust.

**b. In the longwave range. The authors obtain the aerosol properties by Mie calculations as appears in Tables 2 and 4. However, it is not clear what radius are used in Mie calculations. Sometimes the authors assert they use the reff and nevertheless, in table 4 the radii appear in the 2 modes (fine and coarse). Please be clear and consistent.**

The radii values presented in Table 4 are those corresponding to the effective radius of both the fine and the coarse modes. We show these values here because they are directly taken from the different databases. These data are later on processed and converted to the corresponding modal median radius for Mie calculations. Text and corresponding symbols have been modified for clarity purposes.

**3. The results should be analized taking into account the quantitative differences among the aerosol properties used in the aerosol datasets, considering - spectral variation- vertical variation. Considering that the main differences among the aerosol datasets are based on differences in the vertical profile of extinction and absorption, the authors should take into account the work already published in this regard, using other models and different datasets. For example, to help in the interpretation of the differences observed in the shortwave, I would recommend reading of: Meloni et al. 2005. Where the effect of the extinction profile on the calculation of the ARE is analyzed. Different works by Gómez-Amo et al., 2010 and 2011, as well as by Guan et al., 2010. Where the effect of vertically varying the aerosol absorption in the determination of ARE is analyzed.**

We thank the reviewer for these references. Additional discussion considering the results presented by those authors is now included in the manuscript (e.g. Page 17, lines 3-5: *"The vertical distribution of the SSA also influences the radiative fluxes in the SW as demonstrated in previous studies (Gomez-Amo et al., 2010; Guan et al., 2010), contributing to explain the differences observed among the three datasets analyzed here."*)

**Main concerns about results and conclusions sections: SW: Is difficult to understand that with such small differences among the different dataset input (below 1% for radiative fluxes and 0.05 for AOD), why do the authors obtain such large differences in the AREsw (up to 33%)? I think that this is the question you have to answer in this paper, using your data and simulations, which is missing in this paper. At fixed solar zenith angle, the shortwave fluxes are mainly dependent on the AOD. The direct flux is totally driven by the extinction (AOD) and with such small AOD variations between datasets I do not expect large differences in the fluxes (just what you obtain and is shown in Fig. 5 and 6). However, the diffuse radiation is extremely dependent on SSA and the phase function (i.e. asymmetry parameter). If AOD and asymmetry parameter remain fixed, Gómez-Amo et al., (2010) showed that the differences observed in the ARF (at the surface and TOA) are driven by the vertical distribution of SSA that results in different distribution of the diffuse radiation. I would suggest repeating the analysis by removing the small variation of AOD. For example by normalizing the three datasets to the AERONET AOD, or working with the forcing efficiency, and focus the analysis on the variations due to the SSA.**

The AOD difference on June 16 is 0.05, which represents nearly a 22% variation with respect to the AERONET AOD, and therefore is small but still significant, leading to corresponding differences in the ARE. Besides, we have performed a sensitivity study by fixing all the parameters except one as suggested by the reviewer to analyze the influence of each parameter on the final results and we have included the forcing efficiency values. Additional discussion is now included in the manuscript according to these results where appropriate. As for the ARE differences, if we understand the reviewer's point, even though they are large in relative values, absolute values are below 15 $W \cdot m^{-2}$, which is in the same range of the differences obtained for the fluxes. According to the ARE definition, the differences in the ARE obtained values are coherent with the differences in the fluxes.

**I think it would be useful for the interpretation of the results: - to show in Fig. 4 the spectral variation of SSA for the 2 layers observed on June 17, and for the homogenous layer on June 16. - the vertical profiles of SW and LW fluxes in aerosol-free conditions should be shown in Fig. 6 and 9., respectively. (see Meloni et al., 2015; 2018)**

Unfortunately, spectral variation of the SSA is not provided by the aircraft measurements and it is not possible to include it in Fig. 4 for the different layers. The more complete information regarding the vertical distribution of the SSA is already provided by GRASP results in Fig. 3. The aerosol-free conditions profiles are now included in Figs. 6 and 9.

**LW: P20-L20: I do not understand this sentence: "Considering the low influence of the aerosol in the LW radiative fluxes, the influence of the assumed CO2, O3 and the used water vapor profiles and LST are needed to fully explain this discrepancy". Why do you think that the differences in the LW fluxes are due to the assumed CO2, O3 and the used water vapor profiles and LST? Did you change them among the simulations DS1, DS2 and DS3? According to table 4, these values do not change with the dataset considered.**

The reviewer is right, these values did not change for the simulations using DS1, DS2 and DS3. We mean here the differences between GAME retrievals and the aircraft data. The sentence has been modified to make it more clear.

Page 22, lines 21-23: *"Considering the low influence of the aerosol in the LW radiative fluxes, the influence of the assumed $CO_2$, $O_3$ and the used water vapor profiles and LST are needed to fully explain this discrepancy between the aircraft and the simulated profiles."*

**Fig 12. The authors report an ARE offset LW/SW increase with altitude, up 90% at higher altitudes, when there was not aerosol layer anymore. This is totally opposite at what is reported in Meloni et al., (2015), that found the maximum offset at the surface and a negligible variation from the top of the aerosol layer to the TOA. These results should be better discussed and justified.**

A few nuances with regard to the reviewer's comment are necessary to answer this point; there are some similarities between our work and Meloni et al. (2015) but also some differences that need to be considered.

1. We also find that the LW/SW ratio between the highest aerosol layer and the TOA is nearly constant (see Figure 12).
2. In Meloni et al. (2015), SZA = 55.1°. This is comparable to our results on June 17, when SZA = 61.9°. If we look at the evolution of the LW/SW ratio between the surface and the highest aerosol layer (~4 km) on that day a decrease is observed for all datasets (the highest decrease is ~40% for DS3). This decrease in Meloni et al. (2015) is ~50 %.
3. The behavior of the LW/SW ratio plots above the highest aerosol layer may have some intrinsic artefacts due to the different vertical resolutions of the SW and LW model versions. As can be seen in Table 1, the SW version has a 2-km resolution between 2 and 10 km and the LW version has a 1-km resolution below 25 km. This makes that the vertical distribution of the AOD per layer (in the parametrization of GAME) near the top of the aerosol layer is different between

both SW and LW spectral ranges. If we take DS3 on 16 and 17 June, the $ARE_{SW}$ slightly decreases between 6 and 8 km (Fig. 8) while the LW $ARE_{LW}$ is constant above 6 km (Fig. 11). This effect will produce an artificial increase of the LW/SW ratio near or just above the top of the dust layer, which may not be necessarily insignificant as shown on the profile of DS3 on Fig. 12a.

**Minor comments:**
**P2-L3: Please change "contrasted" by "compared"** Done.
**P2-L21: Please rephrase the sentence, its meaning is no clear.** Done.

**P4-L5: Please change "..model estimates sensitivity..." by "..sensitivity of the model estimates.."** Done.
**P5-L2: Please change "..real and imaginary refractive indices..." by "..real and imaginary parts of the refractive index.."** Done.
**P5-L24:Plase remove "particle"** Done.
**P5-L29: Please change "..spatial integral..." by "..vertical integration.."** Done.
**P6-L6: Please change "in" by "by"** Done.
**P9-L27: How the surface temperature was estimated from the measurements at 2m above the ground? please provide a reference.** Done.

The LST is assumed to the temperature measured at the meteorological station, located at 2 m agl. The sentence has been rephrased and discussion is now added regarding this assumption.
Page 19, from line 25: *"On this latter day, larger differences are observed on the Net $F_{LW}$ compared to 16 June, which might be explained by the inaccurate value of LST used due to the lack of precise data. A sensitivity test performed by increasing the air surface temperature measured at the meteorological station 5K indicates that the $^{\uparrow}F_{LW}$ increases its value up to 30 $W \cdot m^{-2}$ at the surface, and around 10 $W \cdot m^{-2}$ from 1 km onwards which is non-negligible. This would lead to an overestimation of the aircraft measured values, but still within a 6% difference. This highlights the need for accurate LST measurements for radiation simulations in the LW spectral range."*

**P11-L12: This sentence is really surprising. I do not understand well the differences in the AOD among the datasets reported in table 4. Since the AOD measured with the CIMEL photometer is imposed as a closure condition in the GRASP and Klett inversions, one would expect similar AOD for DS1 and DS2 datasets, contrarily to what is reported in Table 4. On the other hand, AOD differences are expected from DS1 and DS2, with respect to DS3 (aircraft exctinction). These differences may be also due to the AOD content from surface to the minimun altitude of the aircraft, or for the observation of different airmasess (20km far from ground-based station and aircraft). I think the authors should clearly discuss these AOD differences, since the discussion about the differences in the ARE results is mainly addressed in terms of AOD.**

In our case, the AOD indicated in Table 4 is not exactly the one provided by GRASP, but the integral of the extinction profiles between the surface and the altitude assumed in this work as the top of the aerosol layer, which are the ones used as input for GAME simulations. The GRASP extinction values above this assumed top layer are not null since GRASP assumes stratospheric aerosol presence which exponentially decreases with altitude from the highest altitude used as input (as can be seen in Lopatin et al. (2013),

being the GARRLiC scheme the same used in GRASP lidar retrieval). This fact makes that the integrated GRASP extinction used in this work (integrated until the assumed top aerosol layer but not until TOA) underestimates the AOD from AERONET (stratospheric aerosol is not considered). For the comparison presented here we consider it is more accurate to consider only the region between the surface and the assumed top of the aerosol layer so that we have homogenous criteria for the three datasets. Thus, a larger difference than the uncertainty is obtained between DS1 and DS2.

Page 11, line 24 – Page 12, line 6: *"The AOD values presented here (included in Table 4) are obtained by integrating the $\alpha_{aer}$ profiles at 550 nm from the surface up to the considered top of the aerosol layer (4.3 km on June 16 and 4.7 km on June 17). In GRASP retrieved $\alpha_{aer}$ profiles, values above this top of the aerosol layer are slightly larger than zero since GRASP takes into account stratospheric aerosols by an exponential decay (Lopatin et al., 2013), thus the approach used here to calculate the AOD leads to lower values compared to the column-integrated AOD provided by the sun-photometer. Differences among the three datasets are more noticeable on June 16, when the AOD for DS1 is 0.05 lower than for DS2 and DS3, whereas on June 17 the maximum difference is 0.03, obtained between DS1 and DS2."*

**P13-L2: This approach is similar to the used in Meloni et al., (2015,2018) and Peris-Ferrús et al., (2017). This papers should be cited in this paragraph.**

References have been added.

**P13-L11: Are you sure about this sentence? Is there any typo error? wavelengths below 3 um are not considered LW range. What about for wavelengths over 16 um?**

The Mie code in GAME requires this info to perform the computations of the aerosol properties in the LW. Extrapolation for longer wavelengths is also used, text has been corrected.

**P13-L14: What radius did you use in Mie calculations? the effective radius, or the rF and rC? Table 4, Table 4 caption and the text result confusing. Please be clear.**

See previous response.

**P14-L11. Please change "visible" by "shortwave"** Done

**P14-L12. Please change "thermal" by "longwave"** Done

**P14-L27. When the authors talk about "discrepancies", are they refeering to "relative differences" (Fgame- Faircraft)/Faircraft? The authors should define how they obtain these discrepances, and always use the same term. Sometimes they use "discrepancies" and sometimes"differences".**

Discrepancies here is referred to the absolute difference between the simulated fluxes obtained by the three datasets. Text has been reviewed according to the reviewer's comment to account for possible inconsistencies. The difference with the aircraft data is not considered in this part or the manuscript. In case of relative differences, it is now explicitly indicated in the text and it is defined how they are calculated.

**P15-L5. Please clarify the meaning of this last sentence of the paragraph. The main aerosol effect over the radiative fluxes is due to the AOD, the SSA is a second order effect.**

We agree with the reviewer that this sentence was confusing. It has been rephrased.
Page 16, line 26- Page 17, line 2: *"In our case, the larger AOD assumed for DS2 on both days (see Table 4 and Figure 2), causes the ↓FSW to be slightly lower compared to DS1. For DS3 the AOD is similar to DS2, but the SSA values used, which are relatively smaller compared to those measured by AERONET (see Figure 4), lead to lower values of the radiative fluxes than for DS2."*

**P15-L11. I do not see the influence of the boundary layer due to the distance between the ground-based station and the aircraft leg. The DS3 simulation is set with the aircraft data, so the 20 km distance between the ground-based station and the aircraft leg should be reflected in the results obtained from the DS3 simulation.**

Removed

**P15-L17. The authors should take into account that relative differences about 7% for a Fdownward around 430 and 530 Wmˆ-2 yield absolute differences of 30 and 37 Wmˆ-2. These differences may represent a large fraction of the aerosol effect, with a large contribution to the uncertainties in the determination of ARE using this data. I don't think that these differences are quite insignificat as the authors assert. Have you evaluated them?**

The 7% value indicated in the manuscript correspond to the maximum difference observed, whereas on average differences are below 4%. Considering that the maximum difference observed between the three datasets reaches 19 $W \cdot m^{-2}$ and the uncertainty of the pyranometer is 5 $W \cdot m^{-2,}$ together with the differences between the aircraft and the model in vertical resolution, time sampling and data acquisition and processing techniques a difference of 30 $W \cdot m^{-2}$ between GAME and the aircraft measurements is quite reasonable. Additionally, the uncertainties introduced due to the use of the standard atmosphere or the parameterization of the surface properties may also be partly responsible of the differences observed here. When calculating the ARE, the uncertainties due to the vertical resolution of the model, temporal sampling and the assumption of the standard atmosphere, the gases concentration or surface parameters are minimized.

**P15-L27. Again, a 60 Wmˆ-2 differences between datasets may contribute to large uncertainties in the determination of ARE using this data. Please evaluate this.**

In order to understand the differences observed here an evaluation of the CM11 pyranometer data at the surface against AERONET $^{\downarrow}F_{SW}$ has been performed using the simultaneous data available on June 16 and 17 (6 pairs of data). AERONET surface radiative fluxes have been extensively validated at several different sites around the world by Garcia et al. (2008). In addition, all AERONET sun-photometers are mandatorily calibrated once a year. Large differences are obtained between AERONET and the CM11 pyranometer, reaching up to 130 $W \cdot m^{-2}$ in one of the cases. These results point out to a likely malfunction of the pyranometer during the campaign that would explain the differences observed. We have performed simulations with GAME for the time of the closest AERONET measurement on June 16 (at 16:22UTC), assuming that the aerosol

parameterization is constant with time between the flight time and the photometer measurement (even though there is an AOD increase from 0.23 to 0.25 between 14:30 and 16:22UTC according to the sun-photometer data). This simulation provides $^{\downarrow}F_{SW}$ values at the surface of 564.8, 551.8 and 547.0 W·m$^{-2}$, similar to the 531.4 W·m$^{-2}$ provided by AERONET. On June 17, GAME simulations at 07:40UTC (instead of 07:30UTC, which is the time of the flight), provide values $^{\downarrow}F_{SW}$ at the surface of 466.3, 468.3 and 456.4 W·m$^{-2}$, very close to the AERONET value of 463.7 W·m$^{-2}$.

**P16-L14. I think that the comparison GAME-CERES have no sense, even qualitatively, with CERES overpass 600km. The upward flux at TOA is mainly dependent on surface albedo and clouds, and both can be really different at 600km away. I would suggest remove this comparisons, since they do not present a relevant contribution to this work.**

Removed

**P17-L8:L15. Since this paper is mostly to analyze differences in ARE using different datasets. The authors should carry out a more in deep analysis of the results. Specially regarding to the surface and TOA values. Even if the values obtained for the three datasets fall within the range of the values previosuly observed in the Mediterranean, notice that the values shown in Table 6 differ from a 30-50%. These differences are really large and dependent on the used dataset, and consequently it should be analyzed in detail. Please take into account the following references to help with the interpretation.**

Discussion has been extended taking into account the references suggested by the reviewer.

**P17-L21. Please change "..from Mie calculations from..." by "..from Mie calculations for.."**

Done.

**P17-L23:P18-L4. I think that a more detailed analysis is needed to establish why you observe these differences between GAME and aircraft measurements. On 16 June, the differences may be explained "by the assumed profiles of gases such as CO2, O3 or water vapor, or the uncertainty in the LST", as the authors assert, or may be not. It is just a too simple qualitative explanation.**

For the LW spectral range, the differences are mostly lower than 10 W·m$^{-2}$, which can be considered within the uncertainty limits. Considering the uncertainty of the pyrgeometer, which is 5 W·m$^{-2}$ and the fact that the aircraft and the model present different vertical resolutions and time samplings and the uncertainties due to the use of the standard atmosphere or the parameterization of the surface properties the obtained differences are not significant.

**P18-L7. Please change "the diffuse radiation..." by " the longwave radiation.."**

Done.

**P18-L9. The differences observed are within the uncertainties of the pirgeometers, that for a well mantained and calibrated instrument are below 5 Wm^-2 (Meloni et al., 2012). Therefore, the authors does not need qualitative explanations based on variables as LST to justify the differences.**
Removed.

**P18-L11. As in the SW case, I think it is not worth including the comparison with CERES due to the large distance between the ground-based and satellite measurements.**

Removed.

**P18-L28. Please be consitent, rc or reffc?**
We have updated all the symbols, using $r_{eff,c}$ and $r_{eff,f}$ for clarity.

**Tables 6, 7, 8: The standard deviation is an statistical parameter then it have not sense if obtained over three values only.**

It is included just as an indicator of the variability of the ARF values even though the number of simulations is low, as pointed out by the reviewer.

**REFERENCES:**
**- Meloni, D., A. di Sarra, T. Di Iorio, G. Fiocco, Influence of the vertical profile of Saharan dust on the visible direct radiative forcing, Journal of Quantitative Spectroscopy & Radiative Transfer 93 (2005) 397–413**
**- Meloni, D., Di Biagio, C., di Sarra, A., Monteleone, F., Pace, G., and Sferlazzo, D. M.: Accounting for the solar radiation influence on downward longwave irradiance measurements by pyrgeometers, J. Atmos. Ocean. Technol., 29, 1629–1643, 2012.**
**- Gómez-Amo, J.L., A.diSarra, D.Meloni, M.Cacciani, M.P.Utrillas, Sensitivity of shortwave radiative fluxes to the vertical distribution of aerosol single scattering albedo in the presence of a desert dust layer, Atmospheric Environment 44 (2010) 2787-2791.**
**- Gómez-Amo, J.L., V. Pinti, T. Di Iorio, A. di Sarra, D. Meloni, S. Becagli, V. Bellantone, M. Cacciani, D. Fuà, M.R. Perrone, The June 2007 Saharan dust event in the central Mediterranean: Observations and radiative effects in marine, urban, and sub-urban environments, Atmospheric Environment 45 (2011) 5385-5393**
**- Peris-Ferrús, C., J.L. Gomez-Amo, C. Marcos, M.D. Freile-Aranda, M.P. Utrillas, J.A. Martínez-Lozano, Heating rate profiles and radiative forcing due to a dust storm in the Western Mediterranean using satellite observations, Atmospheric Environment 160 (2017) 142-153.**
**- Guan, H., B. Schmid, A. Bucholtz, and R. Bergstrom, Sensitivity of shortwave radiative flux density, forcing, and heating rate to the aerosol vertical profile, JOURNAL OF GEOPHYSICAL RESEARCH, VOL.115, D06209, doi:10.1029/2009JD012907, 2010**

---

## Author Comment (AC3) · 7 Nov 2018

We would like to thank the reviewers for their efforts and thorough review of our manuscript. We realize that the notes and suggestions made will improve the quality of the paper. Hereafter, the reviewers' comments are presented in bold font and the text included in the manuscript is marked in italics. Line numbering is referred to the new version of the manuscript.

**The paper examines a case of a moderate dust intrusion in Granada with the aim of calculating the dust radiative effect either in the SW and in the LW regions at the surface, at the top of the atmosphere, and within the dust layer by means of the GAME radiative transfer model. The focus of the study is the sensitivity of the SW and LW radiative fluxes and effects on the dust microphysical and optical properties derived in three different ways, using a combination of remote sensing (AERONET and lidar) measurements and the GRASP inversion code, and in situ airborne measurements from the SAFIRE ATR42 aircraft. This study is carried out in the framework of the ChArMEx/ADRIMED campaign and takes advantage of the large observation efforts placed during the project, using either ground-based, airborne and satellite observations. The results of the model calculations are those expected (SW cooling and LW heating by dust, with a not negligible LW/SW ratio), and the case study is not that of an extraordinary dust transport (AOD moderately low). However, the most important conclusion provided by the authors is that optical properties derived from different measurements and with different techniques may provide non-negligible differences in the radiative effects, and should be of concern when estimating dust forcing. I recommend publication, but after some major issues are resolved by the authors.**

Major issues
**The main issue on the presented results concerns the simulation of the SW irradiances using the three different aerosol optical properties DS1, DS2, and DS3, and the evaluation of the ARE. The authors found that very close SW irradiances at surface are obtained with the three datasets (the values seem coincident in Figure 7, but no quantitative information is provided in the text), while differences in the vertical profiles of the downward SW irradiances simulated with GAME are visible in Figure 6 close to the surface when using DS1 or DS2 (which seems to provide identical irradiances than DS3). So there seems to be an inconsistency between the simulations of the vertical profiles and of the surface SW irradiance.**

The same values are represented in both Figures 6 and 7 for the different datasets. They seem inconsistent because of the scales and symbols used. Figures have been modified for clarity. Additionally, quantitative information about Figure 7 is now included in the text.

**As a second point, the LW ARE is very low, due to the values of the AOD, and they might be comparable to the model uncertainties. So a careful evaluation of the uncertainties on the model output should be performed. The fact that the net LW irradiance on 17 June is overestimated by the model is likely not a problem of the**

**CO2 and O3 profiles (the water vapor one should have been taken into account in the simulation, as it measured during the flights) used in the model (to my knowledge the impact of this minor gas is negligible), but comes from the fact that NET=Fâ E̦ Ș-Fâ E̦S, and Fâ E̦ Ș is overestimated, while Fâ E̦S is comparable to measurements. The model overestimation of the LW irradiance at the surface on 17 June (which, however, is as large as the measurement uncertainty) cannot be due to clouds affecting the measurements and not accounted in the model, because clouds increase the LW irradiance, and this should cause a model underestimation. I suggest to perform a sensitivity study to assess the uncertainty on the modelled SW and LW irradiances due to the uncertainty of the input parameters.**

We agree with the reviewer that the uncertainty of the modelled irradiances should be estimated. However, a thorough evaluation of the uncertainty in GAME is out of the scope of the present study and it is worthy a publication itself. Anyway, we performed some sensitivity test to study the influence of the input parameters in the outcome of the model. Variations of the SSA within the uncertainty values considered in AERONET (0.02-0.07 depending on the AOD) show a linear variation of the $ARF_{SW}$ at the BOA reaching up to 10 $W \cdot m^{-2}$ for variations in the SSA of 0.07. In the case of the asymmetry parameter, a variation of 5% which is the uncertainty considered for AERONET data, changes of up to 2 $W \cdot m^{-2}$ are observed in the ARF at the BOA. In the case of the AOD, we observed a maximum variation in the $F_{SW}$ of 6.5 $W \cdot m^{-2}$ (0.7%) at the surface, decreasing with height, for changes in the AOD of up to 0.05, which is the difference we observe between the AOD for DS2 and DS1 on June 16. Considering these results, the total uncertainty associated to the aerosol parameterization is approximately 12 $W \cdot m^{-2}$.

Page 16, line 17 onwards: *"In order to quantify these differences, we performed a sensitivity test by varying the AOD while the other parameters were kept constant. We observed a maximum variation in the $F_{SW}$ of 6.5 $W \cdot m^{-2}$ (0.7%) at the surface, decreasing with height, for changes in the AOD of up to 0.05, which is the difference we observe between the AOD for DS2 and DS1 on June 16. This result partly explains the differences among the three datasets. In addition, a sensitivity test performed by varying exclusively the SSA indicates that more absorbing particles are related to less $^{\downarrow}F_{SW}$ at the surface, namely a variation of 1% is observed at the BOA for a decrease in the SSA of 0.03. The influence of the SSA decreases with height being negligible at the TOA. For the $^{\uparrow}F_{SW}$, a decrease of 0.8% is observed at the BOA if more absorbing particles are present, but in this case the influence at the TOA is larger (2.2%)."*

For the LW, we have introduced a variation in the radius of 10% for the fine mode, for the coarse mode and for both simultaneously. In any case, the differences observed in the radiative fluxes are below 1 $W \cdot m^{-2}$. By assuming an uncertainty in N of 10%, the variations in the fluxes are in the same range, always below 1 $W \cdot m^{-2}$. However, this differences translate in differences in the $ARE_{LW}$ of 1.5 and 0.8 $W \cdot m^{-2}$ at the BOA and the TOA, which is relatively large considering that the obtained values range between 2.5-4.1 and 1.3-2.9 $W \cdot m^{-2}$. As discussed before, variations in the refractive index introduce variations in the ARF at the BOA of up to 1 and 0.6 $W \cdot m^{-2}$ at the TOA and the BOA respectively. Additionally, a strong influence of the LST on the $^{\uparrow}F_{LW}$ is observed, as will be discussed later on.

Page 19, line 25 onwards: *"On this latter day, larger differences are observed on the Net $F_{LW}$ compared to 16 June, which might be explained by the inaccurate value of LST used due to the lack of precise data. A sensitivity test performed by increasing the air surface*

*temperature measured at the meteorological station 5K indicates that the $^\uparrow F_{LW}$ increases its value up to 30 W·m$^{-2}$ at the surface, and around 10 W·m$^{-2}$ from 1 km onwards which is non-negligible. This would lead to an overestimation of the aircraft measured values, but still within a 6% difference. This highlights the need for accurate LST measurements for radiation simulations in the LW spectral range. Additionally, a sensitivity test performed by assuming a 10% uncertainty in the PSD parameters ($r_{eff}$, N and σ) leads to an estimated uncertainty of the $F_{LW}$ retrieved by GAME of around 1.2 W·m$^{-2}$. As stated before, the assumption of the refractive index can also introduce variations as large as 0.8 W·m$^{-2}$. Considering the uncertainty of the pyrgeometer and the fact that the aircraft and the model present different vertical resolutions and time samplings and the uncertainties due to the use of the standard atmosphere or the parameterization of the surface properties the obtained differences are not significant.”*

**As the authors state, the CERES observations are too far in space (600 km) and in time (2 hours) to the surface observations, and a quantitative comparison with the RT model simulations is not possible. I think that the approximate results on the TOA fluxes using CERES data should be removed.**

CERES data and related discussion have been removed from the manuscript.

**Minor issues**
**Introduction: a quantitative description of the SW and LW dust radiative effect in the Mediterranean from previous studies is missing.**

Additional information has been included in the introduction:

Page 2, lines18-25: *“The ARE in the Mediterranean region can be responsible for a strong cooling effect both at the surface (or bottom of the atmosphere, BOA) and the top of the atmosphere (TOA). The so-called forcing efficiency (FE), which is defined as the ratio between the ARE and the AOD, for the SW ranges between -150 and -160 W·m-2 for solar zenith angles (SZA) in the range 50-60° (di Biagio et al., 2009), being able to reach values larger than 200 W·m-2 at the BOA during strong dust events in the Mediterranean region (Gomez-Amo et al., 2011). The LW component accounts for an effect of up to 53% of the SW component and with an opposite sign (di Sarra et al. 2011; Perrone et al., 2012; Meloni et al. 2015).”*

**Page 6, line 9: remove “diffuse” before downward radiative fluxes for the LW.**

Done

**Page 7, line 10: the authors should better explain how measurements corresponding to large pitch and roll aircraft angles are filtered. Moreover, this should be applied not only to downward pyrgeometer measurements, but to either pyranometers and pyrgeometers, both downward and upward-looking. The description of the correction of the ATR42 radiation measurements for the variation in the solar position and in the aircraft attitude during flight applies to pyranometers and not the pyrgeometer measurements, as stated in line 10. The final uncertainty in the airborne SW irradiance profiles is not reported.**

The reviewer is right. This paragraph was referred to both pyranometers' and pyrgeometers' measurements. It has been modified.

**Page 8, lines 23-24: a figure showing the airmass back trajectories may be useful.**

Figure of the air mass trajectories will be added as supplementary material.

[Figure]

**Page 8, line 26: change "profile" with "flight". Do the same at Page 9, line 5.**

Done

**Page 9, lines26-27: the authors mean that, due to the lack in MODIS data of LST the air temperature 2 m a.g.l. have been used as surrogate of the LST? This may lead to a relevant underestimation of the upward LW irradiance. Did the authors verify the differences in air and surface temperature in other cases and the impact of using one or the other in the simulation of the upward LW irradiances?**

We performed a comparison between LST MODIS data and the air temperature at the meteorological station located at Granada experimental station using data corresponding to a 3-month period (May-July 2013). During daytime, the LST is on average 9.8 K larger than the surface air temperature around noon, when the satellites overpasses the station. At nighttime, the LST is 3.95 K lower than the surface air temperature. Considering the LST daily cycle and that the time of the flight was at 07:30UTC, we can expect a difference between the LST and the air temperature, but not as large as 9.8 K. A sensitivity test using the air temperature +5 K (which is the difference observed on June 16 between MODIS and the meteorological station) shows that the $^{\uparrow}F_{LW}$ increases its value up to 30 $W \cdot m^{-2}$ at the surface, and around 10 $W \cdot m^{-2}$ with increasing altitude (see attached figure). The effect on the ARF is however negligible at the BOA and it is 0.3 $W \cdot m^{-2}$ at the TOA.

[Figure]

**Page 13 lines 11-12: maybe the authors mean "extrapolation" instead of "interpolation" and "longer" instead of "shorter".**

Text has been modified according to reviewers' suggestions.

**Page 14 line 11: use "solar" and not "visible".**

Done.

**Page 15 line 2: the sentence "Even though the discrepancies in the AOD are within the uncertainty..." is misleading. DS1 and DS2 use AERONET AOD measurements as input, so both should agree within the measurement uncertainty (±0.01) at 550 nm. But a difference of 0.05 is reported in Table 4. This difference should be addressed and explained.**

This sentence has been modified. In our case, the AOD indicated in Table 4 is not exactly the one provided by GRASP (extinction integral from surface to TOA), but the integral of the extinction profiles between the surface and the assumed as tops of the aerosol layers, which are the ones used as input for GAME simulations. In the extinction profiles provided by GRASP, values above these tops are not exactly equal to zero, because GRASP assumed presence of stratospheric aerosols decreasing exponentially with the altitude (see Lopatin et al. 2013). For the comparison presented here we consider it is more accurate to consider only the region between the surface and the top of the aerosol layer so that we have homogenous criteria for the three datasets. Thus, a larger difference than the uncertainty is obtained between DS1 and DS2, since part of the GRASP extinction profiles (stratospheric aerosol) is not considered in the calculation of AOD.

Page 11, line 24 – Page 12, line 6: *"The AOD values presented here (included in Table 4) are obtained by integrating the $\alpha_{aer}$ profiles at 550 nm from the surface up to the considered top of the aerosol layer (4.3 km on June 16 and 4.7 km on June 17). In GRASP retrieved $\alpha_{aer}$ profiles, values above this top of the aerosol layer are slightly larger than zero since GRASP takes into account stratospheric aerosols by an exponential decay (Lopatin et al., 2013), thus the approach used here to calculate the AOD leads to lower values compared to the column-integrated AOD provided by the sun-photometer.*

*Differences among the three datasets are more noticeable on June 16, when the AOD for DS1 is 0.05 lower than for DS2 and DS3, whereas on June 17 the maximum difference is 0.03, obtained between DS1 and DS2."*

**Page 15 lines 5-7: the spectral variation of the SSA is provided only for the wavelengths at which it has been derived (Figure 4). For example DS3 from aircraft measurements is provided only for one wavelength. This not helps in understanding how optical properties may influence the SW radiation in the whole interval.**

The values presented in Figure 4 correspond to the data obtained from the measurements which are used as input in the radiative transfer model. Unfortunately, the spectral variation of the SSA is not provided by the aircraft measurements. Anyway, this part of the manuscript has been rewritten in order to clarify it.

**Page 15 lines 8-9: this sentence is rather obvious, since the comparison with aircraft radiation fluxes can be done only for the altitudes covered by the ATR42.**

Removed.

**Page 15 line 13: maybe the authors mean "below" instead of "above".**

The reviewer is right; it has been modified.

**Page 15 lines 25-28: the authors should present and discuss absolute and relative differences between measurements, AERONET calculations and GAME simulations. I don't understand the sentence stating that on 16 June the radiation presents large values (than 17 June?), please explain. An overestimation by 6% by GAME is much larger than the irradiance uncertainty. The estimation of the simulated irradiances is necessary in order to understand if such a difference is within the model and measurement respective uncertainties, otherwise it should be commented and possible causes addressed.**

This part of the manuscript has been modified.

**Page 17 lines 8-15: the SZA at which the RTM simulations are performed should be cleared. Are they those corresponding to the middle of the F30 and F31 flights (31.49◦ for 16 June and 61.93◦ for 17 June?). The SW ARE depends on the SZA, so when the authors presents the ARE from previous studies are they sure that such values can be direct compared with theirs? I don't think so, since results from Papadimas et al. (2012) are regional summer means. The same goes for the comparison with the other references (Sicard et al, 2014a,b; Barragan et al., 2017). Moreover, ARE depends on AOD. Thus, when comparing cases with different AOD, the radiative forcing efficiency (ARE per unit AOD) may be the most appropriate quantity.**

The SZA used for the simulations is included in Table 4 (31.49º for 16 June and 61.93º for 17 June) and it does correspond to the middle of the flights. More information about the SZA is now added in the text. When comparing with previous studies the SZA is now taken into account. Moreover, the forcing efficiency is also included to ease the comparison with previous studies and avoid the dependence on the AOD values.

**Page 18, line 7: change "diffuse" with "longwave".**

Done.

**Page 18, line 13: the upward and not the downward component of the LW irradiance depends on LST.**

This part has been removed.

**Page 30, Table 3: albedo in not defined in the LW region! Do authors refer to emissivity? Is it spectrally integrated or for a single wavelength?**

It is the surface LW emissivity product provided by CERES, spectrally integrated in the range 4-100 μm. This information is now included in the manuscript.

**Page 34, Figure 5: a shift is visible in the volume concentration profile of 17 June form GRASP code and aircraft measurements. Can the authors comment on it?**

Details on the comparison between GRASP and the aircraft volume concentration profiles are already discussed in detail in Benavent-Oltra et al. (2017). The differences observed between both profiles are attributed to the different temporal sampling (the aircraft data provide instantaneous values whereas GRASP retrieval is performed using a 30-minutes averaged lidar profile) and the spatial separation (around 20 km distance between the aircraft and the lidar measurements), which can be associated to slight changes in the vertical distribution of the aerosol layers.

**Page 35, Figure 6: The uncertainty range of the radiative fluxes from airborne instrumentation should be added in the plot to help understanding.**

The estimated uncertainty for the airborne measured radiative fluxes is 5 W·m$^{-2}$. Due to the horizontal scale used here to show the data, the uncertainty range is not visible in the plots.

---

## Editor Decision (ED1)

**Comments on the revised version of the ms. acp-2018-700 by M. J. Granados-Muñoz et al., entitled "Impact of mineral dust on shortwave and longwave radiation: evaluation of different vertically-resolved parametrizations in 1-D radiative transfer computations"**

François Dulac, 16 December 2018

First, let me thank you for your revision. I consider that it greatly improved the manuscript. I am suggesting a further small revision because I am not totally satisfied with the way you addressed replies to comments 4 and 5 from reviewer #1 in the revised version itself. My minor comment and a few technical corrections are detailed hereafter.

Minor comment:

-I am not fully satisfied with your reply to reviewer #1 comment 5 on large particles observed in the dust plume further North over the western Mediterranean: I do not think this is the case at the moment but I believe that at least part of your text replies to reviewer #1 comment 4 on the sensitivity to absorbing gases and comment 9 on large particles should be included in the article. I insist that although dust size distribution measurements under balloons drifting from Menorca are somewhat distant from yours, as highlighted in your reply to reviewer #1 comment 5, they were performed during the same dust event (see below my suggestion referring to satellite AODs), and the Lagrangian balloon measurements reported in Renard et al. (2018) shows that the dust particle size distribution seems fairly stable during the long-range transport from Africa to Europe, including the largest mode around 30 µm in diameter that they report. Acknowledging this, and reporting the conclusion of your test that this mode has a weak radiative influence in the LW looks important to me. Regarding this sensitivity test reported in your reply to comment 9, note, however, that the indicated 15% proportion for the third, (coarse) mode should apply to the total mass or volume, but not to the number concentration as stated in your reply (the 3$^{rd}$ mode number contribution is negligible, likely $<10^{-4}$%).

Technical corrections:

-P.9, line 7: "not shown" is no longer correct about back-trajectories; please refer to figure S1.

-P.9, l.10: I suggest making a new sentence after the ref. to Renard et al., replacing ", which […]": "Daily maps of MSG-derived AOD over the Mediterranean from June 15 to 18 during the dust event shown in Figure 4 of Renard et al. (2018) shows the regional extension of the plume over the western Mediterranean region".

-P.14, lines 1-2: "GRASP-derived spectral profiles".

-P.14, l.5: "single-wavelength".

-P.14, l.23: use unbreakable hyphen in units (CTRL+8 in Word).

-P.17, l.16: I suggest "comparable" rather than "similar".

-P.18, l.1: Here again, I think "very similar" is not appropriate; I suggest "close to".

-P.18, l.3: "very similar" would apply here.

-P.19, l.9: remove the article "a" before the 2 numbers in %.

-P.19, line 28: "increasing […] by 5K".

---

## Author Response (AR2)

**Response to the editor:**

Comments on the revised version of the ms. acp-2018-700 by M. J. Granados-Muñoz et al., entitled "Impact of mineral dust on shortwave and longwave radiation: evaluation of different vertically-resolved parametrizations in 1-D radiative transfer computations"

**François Dulac, 16 December 2018**

First, let me thank you for your revision. I consider that it greatly improved the manuscript. I am suggesting a further small revision because I am not totally satisfied with the way you addressed replies to comments 4 and 5 from reviewer #1 in the revised version itself. My minor comment and a few technical corrections are detailed hereafter.

**Minor comment:**

-I am not fully satisfied with your reply to reviewer #1 comment 5 on large particles observed in the dust plume further North over the western Mediterranean: I do not think this is the case at the moment but I believe that at least part of your text replies to reviewer #1 comment 4 on the sensitivity to absorbing gases and comment 9 on large particles should be included in the article. I insist that although dust size distribution measurements under balloons drifting from Menorca are somewhat distant from yours, as highlighted in your reply to reviewer #1 comment 5, they were performed during the same dust event (see below my suggestion referring to satellite AODs), and the Lagrangian balloon measurements reported in Renard et al. (2018) shows that the dust particle size distribution seems fairly stable during the long-range transport from Africa to Europe, including the largest mode around 30 µm in diameter that they report. Acknowledging this, and reporting the conclusion of your test that this mode has a weak radiative influence in the LW looks important to me. Regarding this sensitivity test reported in your reply to comment 9, note, however, that the indicated 15% proportion for the third, (coarse) mode should apply to the total mass or volume, but not to the number concentration as stated in your reply (the 3rd mode number contribution is negligible, likely <10$_{-4}$%).

We would like to thank again the editor and the reviewers. We agree that the manuscript is much improved thanks to the suggested modifications. Considering your comment, we have added the following parts to the manuscript:

*Page 10, line 23: "Variations in the concentration profiles of the main absorbing gases have low impact of the radiative fluxes and the ARE, thus small uncertainty is introduced by this approach. A sensitivity test performed in the present study, varying the $O_3$ profiles up to double concentrations indicates maximum differences of 4 $W \cdot m^{-2}$ in the $F_{SW}$ and 3.6 $W \cdot m^{-2}$ in the case of the $F_{LW}$. For the ARE, differences are negligible (below 0.2 $W \cdot m^{-2}$)."*

*Page 15, line 1: "A third mode at about 30 µm was detected in Renard et al. (2018) for the same dust event using ballon-borne measurements with concentrations up to $10^{-4}$ particles$\cdot cm^{-3}$. However, this*

*giant mode is not considered in our study due to the lack of data above Granada. Considering the relevance of large particles for the $ARE_{LW}$ (i.e. Perrone and Bergamo, 2011; Sicard et al., 2014a,b; Meloni et al., 2018), neglecting this giant mode may contribute to increase the uncertainties in GAME estimations. However, simulations with GAME assuming the presence of a third mode of similar characteristics to the one observed by Renard et al. (2018) indicate that variations in the ARE are negligible in this case (lower than 0.1 W·m$^{-2}$). Even for much higher concentration ($10^{-1}$ particles·cm$^{-3}$) variations of the ARE of just 0.3 W·m$^{-2}$ at the BOA and 0.15 W·m$^{-2}$ at the TOA are obtained.”*

**Technical corrections:**

**-P.9, line 7: "not shown" is no longer correct about back-trajectories; please refer to figure S1.**
Done

**-P.9, l.10: I suggest making a new sentence after the ref. to Renard et al., replacing ", which […]": "Daily maps of MSG-derived AOD over the Mediterranean from June 15 to 18 during the dust event shown in Figure 4 of Renard et al. (2018) shows the regional extension of the plume over the western Mediterranean region".**
Done

**-P.14, lines 1-2: "GRASP-derived spectral profiles".** Done.

**-P.14, l.5: "single-wavelength".** Done.

**-P.14, l.23: use unbreakable hyphen in units (CTRL+8 in Word).** Done.

**-P.17, l.16: I suggest "comparable" rather than "similar".** Done.

**-P.18, l.1: Here again, I think "very similar" is not appropriate; I suggest "close to".** Done.

**-P.18, l.3: "very similar" would apply here.** Done.

**-P.19, l.9: remove the article "a" before the 2 numbers in %.** Done.

**-P.19, line 28: "increasing […] by 5K".** Done.

[revised manuscript text omitted]